# Probabilistic Robust Accuracy is Bounded

## Abstract

Adversarial samples pose a security threat to many critical systems built on neural networks. It has recently been proven that achieving deterministic robustness (*i.e.*, complete elimination of adversarial samples) always comes at an unacceptable cost to accuracy. As a result, probabilistic robustness (where the probability of retaining the same label within a vicinity is at least $1 - \kappa$) has been proposed as a promising compromise. However, existing training methods for probabilistic robustness still experience non-trivial accuracy loss. It remains an open question what the upper limit on accuracy is when optimizing for probabilistic robustness, and whether there is a specific relationship between $\kappa$ and this potential bound. This work studies these problems from a Bayes error perspective. We find that while Bayes uncertainty does affect probabilistic robustness, its impact is smaller than that on deterministic robustness. This reduced Bayes uncertainty allows a higher upper bound on probabilistic robust accuracy than that on deterministic robust accuracy. Further, we show that voting within the vicinity always improves probabilistic robust accuracy and the upper bound of probabilistic robust accuracy monotonically increases as $\kappa$ grows. Our empirical findings also align with our results. This study thus presents a theoretical argument supporting probabilistic robustness as the appropriate target for achieving neural network robustness.

## 1 Introduction

Neural networks (NNs) have achieved remarkable success in various applications, including many security-critical systems (Kurakin et al., 2017b; Sharif et al., 2016). At the same time, several security vulnerabilities in NNs have been identified, including adversarial attacks that generate adversarial samples. Adversarial samples are inputs that are carefully crafted by adding human imperceptible perturbation to normal inputs to trigger wrong predictions (Kurakin et al., 2017a). Their presence is particularly concerning in critical NN applications and they remain a relevant security concern in the era of large models (Yin et al., 2023; Hu et al., 2024; Kumar, 2024).

To defend against adversarial samples, various methods for improving a model's robustness have been proposed. Adversarial training works by training NNs with a mix of normal and adversarial samples, either pre-generated or generated during training. As it does not carry a formal guarantee on the achieved robustness (Zhang et al., 2019b), adversarially trained NNs are potentially vulnerable to new types of adversarial attacks (Liu et al., 2019; Tramer et al., 2020). In contrast, certified training aims to provide a formal guarantee of robustness. Methods in this category typically incorporate robustness verification techniques during training (Xu et al., 2020), *i.e.*, they aim to find a valuation of network parameters such that the model is provably robust with respect to the training samples and some definition of vicinity. However, they are deemed impractical for several reasons, particularly due to the recently proven irreducible errors stemming from Bayes error (the inherent inaccuracies in collecting or labelling training samples), which constrain the maximum achievable accuracy (Chiang et al., 2020; Zhang & Sun, 2024).

Recent studies suggest that probabilistic robustness, defined as the probability of adversarial samples within a neighbourhood being no greater than a specified tolerance level $\kappa$ (*e.g.*, 0.1%), may be sufficient for many practical applications (Robey et al., 2022; Li et al., 2022; Zhang et al., 2023b). Furthermore, it is shown to be achievable with a smaller accuracy drop ($\sim$5% when $\kappa = 0.1$) and significantly reduced computational cost compared to certified training methods (Robey et al., 2022). Probabilistic robustness thus offers a balance between ensuring strong security and preserving accuracy. However, it remains an open question how Bayes errors similarly limit the achievable

performance when optimizing for probabilistic robustness, measured in terms of probabilistic robust accuracy. Furthermore, if an upper limit does exist, how is it related to the tolerance level $\kappa$? Answering this question would offer practical guidance on balancing robustness and accuracy in real-world applications.

In this work, we aim to answer these questions. The Bayes error, in the context of statistics and machine learning, is a fundamental concept related to the inherent uncertainty in any classification system (Ishida et al., 2023). It represents the minimum error for any classifier on a given problem and is determined by the overlap in the probability distributions of different classes (Fukunaga, 1990). We remark that the relevance of Bayes error in simple classification tasks may occasionally be questioned given that many datasets, such as MNIST, provide a single, definite label for each input (LeCun et al., 1998). However, real-world data often lacks this clarity due to inevitable information loss, *e.g.*, during image capture or compression. For instance, the CIFAR-10H dataset showcases that over a third of CIFAR-10 inputs can be re-annotated with uncertain labels by human annotators (CIFAR-10H) (Peterson et al., 2019). Thus, this uncertainty leads to Bayes errors, which fundamentally constrain not only vanilla accuracy (Ishida et al., 2023) but also deterministic robust accuracy (Zhang & Sun, 2024) and, as we show in this work, probabilistic robust accuracy.

We study the limit on the probabilistic robust accuracy resulting from Bayes error. We first derive an optimal decision rule that maximises probabilistic robust accuracy. Similar to the Bayes classifier (Fukunaga & Hostetler, 1975), the optimal decision rule for probabilistic robustness is also a Maximum A Posteriori (MAP (Bassett & Deride, 2019)) probability decision, except that the posterior is regarding the vicinity, not a single input. Then, we show that the error from this optimal decision rule regarding probabilistic robustness is lower bounded by the Bayes error of deterministic robustness, but within a significantly smaller vicinity. After that, a relationship is established between the upper bound of probabilistic robust accuracy and the upper bound of vanilla accuracy or deterministic robust accuracy. We further show that the bound monotonically increases as $\kappa$ grows. Empirically, we show that our bounds are consistent with what is observed on those probabilistically robust neural networks trained on various distributions. In practical terms, our result establishes a significantly higher upper bound on probabilistic robust accuracy compared to deterministic robust accuracy, even when $\kappa$ is very small (*e.g.*, fewer than 1 adversarial sample per 1000). This provides a theoretical basis for endorsing probabilistic robustness as the ideal target for neural network robustness.

## 2 PRELIMINARY AND PROBLEM DEFINITION

This section first reviews the background of robustness in machine learning. Then, we recall the Bayes error for the deterministic robustness of classification. Finally, we define our research problem.

### 2.1 ROBUSTNESS IN NEURAL NETWORK CLASSIFICATION

We put the context in a $K$-class classification problem where a classifier $h : \boldsymbol{x} \mapsto y$ learns to fit a joint distribution $D$ over input space $\mathbb{R}^n$ and label space $\{0, 1, ..., K-1\}$. Let $h(\boldsymbol{x}) \in \{0, 1, ..., K-1\}$ denote prediction, and an error captures the difference between $h(\boldsymbol{x})$ and $y$. That is, vanilla accuracy is $\Upsilon_{\text{acc}}^+(D, h) = \mathrm{E}_{(\mathbf{x}, \mathbf{y}) \sim D}\left[\mathbf{1}_{h(\mathbf{x})=\mathbf{y}}\right]$ and thus its error is $\Upsilon_{\text{acc}}^-(D, h) = 1 - \Upsilon_{\text{acc}}^+(D, h)$. The capital Upsilon with a plus sign denotes accuracy itself, while a minus sign denotes its corresponding error.

Robustness $\Upsilon_{\text{rob}}^+(D, h, \mathbb{V})$ measures the change in prediction when a perturbation occurs on the input (Szegedy et al., 2014). If the prediction changes when an input is perturbed, then this input is an adversarial example. Formally, an input $\boldsymbol{x}'$ is an adversarial example of an input-label pair $(\boldsymbol{x}, y)$ if $\left(h(\boldsymbol{x}) = y\right) \wedge \left(h(\boldsymbol{x}') \neq h(\boldsymbol{x})\right) \wedge \left(\boldsymbol{x}' \in \mathbb{V}(\boldsymbol{x})\right)$ (Goodfellow et al., 2015; Kurakin et al., 2017a), where $\mathbb{V}(\boldsymbol{x})$ is the vicinity at $\boldsymbol{x}$. We define robustness to be the probability of *not* observing an adversarial example (Lin et al., 2019), as defined in Equation (1).

$$\Upsilon_{\text{rob}}^+(D, h, \mathbb{V}) = P_{(\mathbf{x}, \mathbf{y}) \sim D}\left(\left(h(\mathbf{x}) = \mathbf{y}\right) \wedge \forall \boldsymbol{x}' \in \mathbb{V}(\mathbf{x}). \, h(\boldsymbol{x}') = h(\mathbf{x})\right) \qquad (1)$$

*Remark* 2.1. $\boldsymbol{x}$-vicinity can also be equivalently expressed in a distribution notation $\mathcal{V}(\boldsymbol{x})$, which is defined over the $\mathbb{X}$, and $\mathbb{X} \subseteq \mathbb{R}^n$. To represent vicinity, $\mathcal{V}$ has a probability density function (PDF) $v$ that takes value 0 for every input not in $\mathbb{V}$. Their equivalence and derivation is shown as follows.

**Vicinity**  A vicinity is a set of points around a particular input, *i.e.*, for any input $\boldsymbol{x} \in \mathbb{X}$, the vicinity of $\boldsymbol{x}$ is written as $\mathbb{V}(\boldsymbol{x})$. Yet, there are more convenient forms to express the characteristics of a point in some vicinity (Zhang & Sun, 2024).

Given $\boldsymbol{x} \in \mathbb{X}$, consider a probabilistic distribution that may only generate outcomes from $\mathbb{V}(\boldsymbol{x})$. We denote this distribution as $\mathcal{V}(\boldsymbol{x})$. Thus, any possible outcome drawn from $\mathcal{V}(\boldsymbol{x})$ is in $\boldsymbol{x}$-vicinity, and vice versa. Formally, let $(v : \mathbb{X} \to \mathbb{R})$ denote PDF of $\mathcal{V}$, and we have $\forall \boldsymbol{x}' \in \mathbb{X}, v(\boldsymbol{x}' - \boldsymbol{x}) > 0 \iff \boldsymbol{x}'$ is in $\boldsymbol{x}$-vicinity. A typical form of $v$ (or $\mathcal{V}$) is a uniform distribution. In this case, a vicinity function can be defined as Equation (2).

$$v_{\boldsymbol{x}}(\boldsymbol{x}') = \begin{cases} \left( \int_{\mathbb{V}(\boldsymbol{x})} d\boldsymbol{x}'' \right)^{-1}, & \text{if } \boldsymbol{x}' \in \mathbb{V}(\boldsymbol{x}) \\ 0, & \text{otherwise} \end{cases} \tag{2}$$

Now we shift the x-coordinate of Equation (2) by $+\boldsymbol{x}$, we get

$$v_{\boldsymbol{0}}(\boldsymbol{x}' - \boldsymbol{x}) = \begin{cases} \left( \int_{\mathbb{V}(\boldsymbol{0})} d\boldsymbol{x}'' \right)^{-1}, & \text{if } \boldsymbol{x}' - \boldsymbol{x} \in \mathbb{V}(\boldsymbol{0}) \\ 0, & \text{otherwise} \end{cases} \tag{3}$$

We assume the given vicinity scheme is translation invariant, and then we can drop the subscript $\boldsymbol{0}$ of $v_{\boldsymbol{0}}$. We also let $\epsilon_{\mathrm{v}} \equiv \int_{\mathbb{V}(\boldsymbol{0})} d\boldsymbol{x}''$ represent the size of the vicinity, which is constant. Additionally, we use a generic input $\boldsymbol{x} \in \mathbb{X}$, rather than the bound centre input, as the variable notation of $v$. Thus, the vicinity function $v : \mathbb{X} \to \left\{ 0, \epsilon_{\mathrm{v}}^{-1} \right\}$ can be expressed as Equation (4), and an example of a one-dimensional input's vicinity is shown in Figure 4.

$$v(\boldsymbol{x}) = \begin{cases} \epsilon_{\mathrm{v}}^{-1} & \text{if } \boldsymbol{x} \in \mathbb{V}(\boldsymbol{0}) \\ 0, & \text{otherwise} \end{cases} \tag{4}$$

In summary, the following four statements are equivalent: (1) $\boldsymbol{x}' \in \mathbb{V}(\boldsymbol{x})$; (2) $v(\boldsymbol{x}' - \boldsymbol{x}) > 0$; (3) an $\boldsymbol{x}'$ can be drawn from $\mathcal{V}(\boldsymbol{x})$; (4) $\boldsymbol{x}'$ is a neighbour of $\boldsymbol{x}$. Appendix A.1 provides a detailed derivation of the vicinity function notation.

**Deterministic Robustness**  Deterministic robustness requires a zero probability of adversarial samples occurring in a vicinity. It is difficult as achieving $\forall \boldsymbol{x}' \in \mathbb{V}(\mathbf{x}), h(\boldsymbol{x}') = h(\mathbf{x})$ is challenging. Although adversarial training (Goodfellow et al., 2015) empirically reduces adversarial samples (Ganin et al., 2016), it lacks a formal guarantee. Meanwhile, certified training (Müller et al., 2022) guarantees deterministic robustness through integrating NN verification during training, often resulting in a significant accuracy drop (35%) (Li et al., 2023).

**Probabilistic Robustness**  While deterministic robustness is often infeasible without seriously compromising accuracy, probabilistic robustness claims to balance robustness and accuracy (Robey et al., 2022). Probabilistic robustness is defined as in Equation (5), where a tolerance level $\kappa$ limits the probability of having adversarial samples in a vicinity $\mathcal{V}$. Here, a small portion (such as 1% (Zhang et al., 2023b; Robey et al., 2022)) of adversarial samples within the vicinity is considered acceptable. Probabilistic robustness is often sufficient in practice (Robey et al., 2022). Indeed, safety certification of many safety-critical domains such as aviation requires keeping safety violation probabilities below a non-zero threshold (Guerin et al., 2021).

$$\Upsilon_{\mathrm{prob}}^{+}(D, h, \mathcal{V}, \kappa) = P_{(\mathbf{x}, \mathbf{y}) \sim D}\Big( \big( h(\mathbf{x}) = \mathbf{y} \big) \wedge \big( \big( P_{\mathbf{x}' \sim \mathcal{V}(\mathbf{x})}(h(\mathbf{x}') \neq h(\mathbf{x})) \leq \kappa \big) \Big) \tag{5}$$

## 2.2 AN UPPER BOUND OF DETERMINISTIC ROBUSTNESS FROM BAYES ERROR

**The Bayes Error**  In the presence of uncertainty in data distribution, a classifier (no matter how it is trained) inevitably makes some wrong predictions. Bayes error quantifies this inherent uncertainty and represents the irreducible error in accuracy (Fukunaga, 1990; Garber & Djouadi, 1988; Ripley, 1996), formally captured in Equation (6).

$$\min_{h \in \{ \mathbb{R}^n \to \{ 0, 1, \dots, K-1 \} \}} \Upsilon_{\mathrm{acc}}^{-}(D, h) = \mathrm{E}_{(\mathbf{x}, \mathbf{y}) \sim D} \left[ 1 - \max_k p(\mathbf{y} = k | \mathbf{x}) \right] \tag{6}$$

A classifier achieves the Bayes error when its predictions correspond to the class with maximal posterior probability. Such a classifier is known as a Bayes classifier. The posterior of other classes thus contributes to the irreducible error. An example illustrating the Bayes error is shown in Figure 1a.

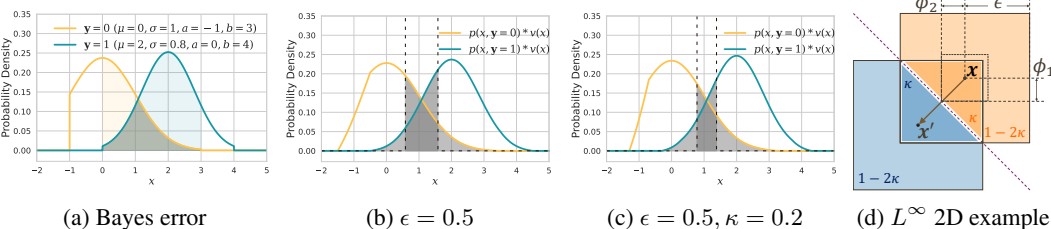

(a) Bayes error      (b) $\epsilon = 0.5$      (c) $\epsilon = 0.5, \kappa = 0.2$      (d) $L^\infty$ 2D example

Figure 1: Two truncated normal distributions are used to visualise the Bayes error of (a) vanilla accuracy, (b) deterministic robust accuracy and (c) probabilistic robust accuracy. (d) Example of Corollary 3.6. The nearest adversarial example of $\boldsymbol{x}$ is at the midpoint of $\boldsymbol{x}$ and $\boldsymbol{x'}$. Both $\boldsymbol{x}$ and $\boldsymbol{x'}$ are probabilistically robust but $h(\boldsymbol{x}) \neq h(\boldsymbol{x'})$. The dashed box with side length $2\phi_i$ representes $\mathbb{V}^{\downarrow\kappa}(\boldsymbol{x})$.

**Bayes Error for Deterministic Robustness**    Prior work (Zhang & Sun, 2024) shows that optimising towards deterministic robustness makes the Bayes error worse. Besides the posterior of other classes forcing a prediction to be consistent with its neighbours constitutes another source of Bayes error (Zhang & Sun, 2024). As in Equation (7), the Bayes error for deterministic robustness can be derived from the Bayes error of a convolved distribution $D' = D * v$. In $D'$, $p(\mathbf{x}, \mathbf{y})$ is convolved from vicinity $v$ and $p(\mathbf{x}, \mathbf{y})$ in $D$.

$$\min_{h \in \{\mathbb{R}^n \to \{0,1,...,K-1\}\}} \Upsilon_{\text{rob}}^-(D, h, \mathbb{V}) = \mathrm{E}_{(\mathbf{x},\mathbf{y}) \sim D'} \left[ 1 - \max_k p(\mathbf{y} = k | \mathbf{x}) \mathbf{1}_{\mathbf{x} \notin \mathbb{K}_{D^\dagger}} \right] \quad (7)$$

where $\Upsilon_{\text{rob}}^-(D, h, \mathbb{V}) = 1 - \Upsilon_{\text{rob}}^+(D, h, \mathbb{V})$. $D^\dagger = \lceil D' \rceil * v$ where $\lceil D' \rceil$ is the "hardened" distribution of $D'$, *i.e.*, one-hot of Argmax posterior. $\mathbb{K}_{D^\dagger} = \{ \boldsymbol{x} \mid (\mathbf{x}, \mathbf{y}) \sim D^\dagger, \max_k p(k|\mathbf{x} = \boldsymbol{x}) < 1 \}$ represents a domain near the boundary where the marginal probability rather than joint probability contributes to the Bayes error for deterministic robustness. Therefore, the Bayes error for deterministic robustness of $D$ is the Bayes error of $D'$ plus the joint probability of non-max classes in $\mathbb{K}_{D^\dagger}$. As shown in (Zhang & Sun, 2024), deterministic robustness $\Upsilon_{\text{rob}}^+(D, h, \mathbb{V})$ has an upper bound of 1 minus this irreducible error. Figure 1b illustrates the Bayes error for deterministic robustness.

## 2.3   Problem Definition

The primary focus of this study is finding an upper bound of probabilistic robust accuracy. Further, we aim to establish a relation between this upper bound and the tolerance level, *i.e.*, $\kappa$. Formally, we solve the minimisation problem in Equation (8) where $\Upsilon_{\text{prob}}^-(D, h, \mathcal{V}, \kappa) = 1 - \Upsilon_{\text{prob}}^+(D, h, \mathcal{V}, \kappa)$.

$$\min_{h \in \{\mathbb{R}^n \to \{0,1,...,K-1\}\}} \Upsilon_{\text{prob}}^-(D, h, \mathcal{V}, \kappa) \quad (8)$$

## 3   Method

In the following, we study the upper bound of probabilistic robust accuracy. We first model the error when optimising towards probabilistic robustness and derive an optimal decision rule. Then, we study the Bayes error obtained from this rule. Further, we formally establish the relationship between the upper bounds of vanilla accuracy, probabilistic robust accuracy, and deterministic robust accuracy.

### 3.1   Error Modelling and Optimal Decision Rule for Probabilistic Robustness

To find Bayes error when optimising towards probabilistic robustness given distribution $(\mathbf{x}, \mathbf{y}) \sim D$, we first model $\Upsilon_{\text{Prob}}^-(D, h, \mathcal{V}, \kappa)$. Intuitively, an error happens if the prediction is wrong or many of the samples in the vicinity are predicted wrongly. We denote the error from the former case as incorrectness and the latter as inconsistency. We analyze each type of error and their combined effect.

**Incorrectness**    The incorrectness for any example $(\boldsymbol{x}, y)$ is simply $\mathbf{1}_{h(\boldsymbol{x}) \neq y}$. The incorrectness of a prediction at an input $\boldsymbol{x}$ given all possible labels $y \in \{0, 1, ..., K-1\}$ considers posterior at $\boldsymbol{x}$, as in

Equation (9). Incorrectness (cor) is minimum when $h(\boldsymbol{x})$ equals the class with the highest posterior.

$$e_{\text{cor}}(\boldsymbol{x}, h; P(\mathbf{y} \mid \boldsymbol{x})) = \sum_{y=0}^{K-1} P(\mathbf{y} = y \mid \mathbf{x} = \boldsymbol{x}) \mathbf{1}_{h(\boldsymbol{x}) \neq y} = 1 - \sum_{y=0}^{K-1} P(\mathbf{y} = y \mid \mathbf{x} = \boldsymbol{x}) \mathbf{1}_{h(\boldsymbol{x}) = y} \quad (9)$$

**Inconsistency** Inconsistency results from prediction at $\boldsymbol{x}$ being not the same as some of its neighbours. Let $P_{\mathbf{t} \sim \mathcal{V}(\boldsymbol{x})}\big(h(\mathbf{t}) \neq h(\boldsymbol{x})\big)$ denote the probability of a neighbour of $\boldsymbol{x}$ having a prediction different from $\boldsymbol{x}$. Since this probability is parameterised by $h(\boldsymbol{x}) \in \{0, 1, ..., K-1\}$, it can be reformulated as $\sum_{k=0}^{K-1} \mathbf{1}_{h(\boldsymbol{x})=k} P_{\mathbf{t} \sim \mathcal{V}(\boldsymbol{x})}\big(h(\mathbf{t}) \neq k\big)$. Let $\mu_k(\boldsymbol{x}) \triangleq P_{\mathbf{t} \sim \mathcal{V}(\boldsymbol{x})}\big(h(\mathbf{t}) = k\big)$ and $\sum_{k=0}^{K-1} \mu_k(\boldsymbol{x}) = 1$. Intuitively, $\mu_k$ is the probability of a neighbour predicted as class-$k$. Thus, the probability of a neighbour of $\boldsymbol{x}$ having a different prediction from $\boldsymbol{x}$ can be written as Equation (10).

$$P_{\mathbf{t} \sim \mathcal{V}(\boldsymbol{x})}\big(h(\mathbf{t}) \neq h(\boldsymbol{x})\big) = \sum_{k=0}^{K-1} \mathbf{1}_{h(\boldsymbol{x})=k}\big(1 - \mu_k(\boldsymbol{x})\big) = 1 - \sum_{k=0}^{K-1} \mu_k(\boldsymbol{x}) \mathbf{1}_{h(\boldsymbol{x})=k} \quad (10)$$

Inconsistency exists when $P_{\mathbf{t} \sim \mathcal{V}(\boldsymbol{x})}\big(h(\mathbf{t}) \neq h(\boldsymbol{x})\big) > \kappa$. This thresholding can be represented by a unit step function ($u$) that takes an input $P_{\mathbf{t} \sim \mathcal{V}(\boldsymbol{x})}\big(h(\mathbf{t}) \neq h(\boldsymbol{x})\big) - \kappa$. Thus, inconsistency (cns) at $\boldsymbol{x}$ is expressed as Equation (11). Also, Lemma 3.1 suggests that $\kappa$ takes value from $[0, 1/2)$.

$$e_{\text{cns}}(\boldsymbol{x}, h; \mathcal{V}, \kappa) = u\left(P_{\mathbf{t} \sim \mathcal{V}(\boldsymbol{x})}\big(h(\mathbf{t}) \neq h(\boldsymbol{x})\big) - \kappa\right) = u\left(1 - \kappa - \sum_{k=0}^{K-1} \mu_k(\boldsymbol{x}) \mathbf{1}_{h(\boldsymbol{x})=k}\right) \quad (11)$$

**Lemma 3.1.** *For the prediction of input $\boldsymbol{x}$ to be consistent, at most one class has a prediction probability $\geq 1 - \kappa$ in $\boldsymbol{x}$-vicinity. Thus, $\kappa < \frac{1}{2}$. (Proof is provided in Appendix B.1.)*

**The overall error across the distribution** Considering probabilistic robustness, the error at input $\boldsymbol{x}$ is a combined error of $e_{\text{cor}}$ and $e_{\text{cns}}$ at $\boldsymbol{x}$. We need two intuitions to derive the combined error. First, if $e_{\text{cns}}(\boldsymbol{x}, h; \mathcal{V}, \kappa) = 1$, the combined error is always 1. Second, if $e_{\text{cns}}(\boldsymbol{x}, h; \mathcal{V}, \kappa) = 0$, the combined error equals $e_{\text{cor}}(\boldsymbol{x}, h; P(\mathbf{y}|\boldsymbol{x}))$. Note that $e_{\text{cor}}$ takes value from $[0, 1]$ and $e_{\text{cns}}$ takes binary value from $\{0, 1\}$. The combined error $e$ is expressed as Equation (12) whose derivation is in Appendix B.2.

$$e(\boldsymbol{x}, h; P(\mathbf{y}|\boldsymbol{x}), \mathcal{V}, \kappa) = (1 - e_{\text{cns}}(\boldsymbol{x}, h; \mathcal{V}, \kappa))e_{\text{cor}}(\boldsymbol{x}, h; P(\mathbf{y} \mid \boldsymbol{x})) + e_{\text{cns}}(\boldsymbol{x}, h; \mathcal{V}, \kappa)$$

$$= 1 - u\left(\kappa - 1 + \sum_{k=0}^{K-1} \mu_k(\boldsymbol{x}) \mathbf{1}_{h(\boldsymbol{x})=k}\right)\left(\sum_{y=0}^{K-1} P(\mathbf{y} = y \mid \mathbf{x} = \boldsymbol{x}) \mathbf{1}_{h(\boldsymbol{x})=y}\right) \quad (12)$$

Note that in general, the errors are functions of $\boldsymbol{x}$, $h$, $D$, $\mathcal{V}$, and $\kappa$. For simplicity, when $h$, $D$, $\mathcal{V}$, or $\kappa$ can be inferred from the context, we simply omit them, *e.g.*, the simplest case is written as $e_{\text{cor}}(\boldsymbol{x})$ to denote the incorrectness, $e_{\text{cns}}(\boldsymbol{x})$ to denote inconsistency, and $e(\boldsymbol{x})$ to denote the combined error.

To model the distribution-wise error of classifier $h$ on $(\mathbf{x}, \mathbf{y}) \sim D$, we compute the expectation of $e(\boldsymbol{x})$ across $D$. Formally, $\Upsilon_{\text{prob}}^-(D, h, \mathcal{V}, \kappa) = \int_{\boldsymbol{x} \in \mathbb{R}^n} e(\boldsymbol{x}) p(\mathbf{x} = \boldsymbol{x}) d\boldsymbol{x}$, where $p(\mathbf{x})$ is the marginal probability in $D$. Hereby, we get $\Upsilon_{\text{prob}}^-$, the error when optimising towards probabilistic robustness.

To minimise $\Upsilon_{\text{prob}}^-$ of any measurable classification function $h$, we explore the optimal decision rules for probabilistic robustness. From Equation (12), we can establish a Maximum A Posteriori optimal decision rule, whose formal statement is given in Theorem 3.2.

**Theorem 3.2.** *If $h^*$ is optimal for the probabilistic robustness on a given distribution, i.e., $h^* = \arg\min_h \int_{\boldsymbol{x} \in \mathbb{R}^n} e(\boldsymbol{x}) p(\mathbf{x} = \boldsymbol{x}) d\boldsymbol{x}$, we would always have $\forall \boldsymbol{x} \in \mathbb{R}^n, h^*(\boldsymbol{x}) = \arg\max_k \mu_k(\boldsymbol{x})$.*

*Proof.* Let $h_1$ and $h_2$ be two distinct classification functions such that $h_1(\boldsymbol{x}) = \arg\max_k \mu_k(\boldsymbol{x})$ and $h_2(\boldsymbol{x}) \neq h_1(\boldsymbol{x})$. If we can prove $e(\boldsymbol{x}, h_1) \leq e(\boldsymbol{x}, h_2)$, then we can know $h_1$ must be optimal for probabilistic robustness. First, we denote $k_1 = h_1(\boldsymbol{x})$ and $k_2 = h_2(\boldsymbol{x}) \neq k_1$. Then,

$$e(\boldsymbol{x}, h_1) - e(\boldsymbol{x}, h_2) = u\big(\kappa - 1 + \mu_{k_2}(\boldsymbol{x})\big) P(\mathbf{y} = k_2 | \mathbf{x} = \boldsymbol{x}) - u\big(\kappa - 1 + \mu_{k_1}(\boldsymbol{x})\big) P(\mathbf{y} = k_1 | \mathbf{x} = \boldsymbol{x}). \quad (13)$$

Since $\mu_{k_2}(\boldsymbol{x}) \leq \mu_{k_1}(\boldsymbol{x})$, we get $\mu_{k_2}(\boldsymbol{x}) \leq 1/2$. Recall $\kappa < 1/2$ from Lemma 3.1, we get $\kappa - 1 + \mu_{k_2}(\boldsymbol{x}) < 1/2 - 1 + 1/2 = 0$. Consequently, we have $u\big(\kappa - 1 + \mu_{k_2}(\boldsymbol{x})\big) = 0$. Therefore, $e(\boldsymbol{x}, h_1) - e(\boldsymbol{x}, h_2) = -u\big(\kappa - 1 + \mu_{k_1}(\boldsymbol{x})\big) P(\mathbf{y} = k_1 | \mathbf{x} = \boldsymbol{x}) \leq 0$. This inequality applies to any input $\boldsymbol{x}$. Hence, a classification function like $h_1$ is optimal. An extended proof is in Appendix B.3. $\square$

Intuitively, the theorem states that when optimising towards probabilistic robustness, a Bayes (optimal) classifier would always classify a sample with the most popular label in the vicinity.

## 3.2 BAYES ERROR FOR PROBABILISTIC ROBUSTNESS FROM BAYES ERROR FOR DETERMINISTIC ROBUSTNESS

The Bayes classifier, regarding probabilistic robustness, is closely related to the most popular label in its vicinity, leading us to study the properties of $\mu_k$. Intuitively, $\mu_k$ is the probability of a neighbour predicted as class-$k$. Formally, $\mu_k$ has an equivalent convolutional form as

$$\mu_k(\boldsymbol{x}) = P_{\mathbf{t} \sim \mathcal{V}(\boldsymbol{x})}\big(h(\mathbf{t}) = k\big) = \int_{\boldsymbol{t} \in \mathbb{R}^n} \mathbf{1}_{h(\boldsymbol{t})=k}\, v(\boldsymbol{x} - \boldsymbol{t})\, d\boldsymbol{t} = (\mathbf{1}_{h(\cdot)=k} * v)(\boldsymbol{x}), \qquad (14)$$

where $*$ denotes convolution and $\mathbf{1}_{h(\cdot)=k}$ denotes an indicator function returning 1 if $h$ of input equals $k$. $v$ is the probability density function of vicinity distribution, *e.g.*, uniform distribution.

Intuitively, convolution acts as a smoothing operation. Thus, $\mu_k(\boldsymbol{x})$ is expected to change *gradually* as $\boldsymbol{x}$ moves in $\mathbb{R}^n$. Similarly, $\arg\max_k \mu_k(\boldsymbol{x})$ is unlikely to switch frequently. This implies that under the optimal probabilistic robustness condition, predictions do not change randomly or frequently (in $\mathbb{R}^n$) but exhibit a form of continuity. Lemma 3.3 and Theorem 3.4 formally states this intuition. Specifically, Lemma 3.3 states that $\mu_k(\boldsymbol{x})$ changes *gradually* as $\boldsymbol{x}$ moves in $\mathbb{R}^n$. Moreover, Theorem 3.4 states that the Bayes classifier (for probabilistic robustness) achieves deterministic robustness with a much smaller vicinity at any input that achieves probabilistic robustness.

**Lemma 3.3.** *The change in $\mu_k$ resulting from shifting an input by a certain distance $\phi$ within the vicinity is bounded in any direction $\hat{\boldsymbol{\phi}}$. Formally: where $\mathbb{S}^{n-1}$ is the set of all unit vectors in $\mathbb{R}^n$,*

$$\forall \boldsymbol{x} \in \mathbb{R}^n, \forall \phi \in \mathbb{R}, \left( \left( \forall \hat{\boldsymbol{\phi}} \in \mathbb{S}^{n-1},\ v\left(\frac{\phi}{2}\hat{\boldsymbol{\phi}}\right) > 0 \right) \to \right.$$

$$\left. \forall \hat{\boldsymbol{\phi}} \in \mathbb{S}^{n-1},\ \left( \left| \mu_k(\boldsymbol{x} + \phi\hat{\boldsymbol{\phi}}) - \mu_k(\boldsymbol{x}) \right| \le 1 - \min_{\hat{\boldsymbol{\phi}}' \in \mathbb{S}^{n-1}} \int_{\boldsymbol{t} \in \mathbb{R}^n} \min\left( v(\boldsymbol{t} - \phi\hat{\boldsymbol{\phi}}'), v(\boldsymbol{t}) \right) d\boldsymbol{t} \right) \right).$$
$$(15)$$

Proof of Lemma 3.3 is given in Appendix B.4. Essentially, for all inputs shifting a distance $\phi$, the $\mu_k$ value difference between the original input and the shifted input will be bounded by the 1 minus the minimum overlap between two vicinities that are $\phi$ apart. From Inequality (15), the correlation between the distance $\phi$ and maximum change of $\mu_k$ can be modelled as a function of $\phi$ expressed as

$$\underset{\max}{\Delta}\, \mu_k(\phi) = \max_{\hat{\boldsymbol{\phi}} \in \mathbb{S}^{n-1}, \boldsymbol{x} \in \mathbb{R}^n} \left| \mu_k(\boldsymbol{x}) - \mu_k(\boldsymbol{x} + \phi\hat{\boldsymbol{\phi}}) \right| \qquad (16)$$

Note that $\Delta_{\max}\mu_k(\phi)$ is a monotonic function, *i.e.*, a greater shift distance $\phi$ is required if the maximal change in $\mu_k$ needs to be increased. Theorem 3.4 leverages this monotonicity to formally show the connection between the Bayes error when optimising towards probabilistic robustness and that towards deterministic robustness.

**Theorem 3.4.** *If $h^*$ is optimal for the probabilistic robustness on a given distribution, i.e., $h^* = \arg\min_{h \in \{\mathbb{R}^n \to \{0,1,...,K-1\}\}} \Upsilon_{\text{prob}}^-(D, h, \mathcal{V}, \kappa)$, then there is a lower bound on the distance between an input $\boldsymbol{x}$ and any of its adversarial samples if probabilistic consistency is satisfied on $\boldsymbol{x}$. Formally,*

$$\forall \boldsymbol{x} \in \mathbb{R}^n. \left( \left( \exists k \in \{0,1,...,K-1\}.\ \mu_k(\boldsymbol{x}) > 1 - \kappa \right) \to \right.$$

$$\left. \forall \boldsymbol{x}' \in \mathbb{R}^n. \left( (h^*(\boldsymbol{x}') = k) \vee \left( |\boldsymbol{x} - \boldsymbol{x}'| \ge (\Delta_{\max}\mu_k)^{-1}(1/2 - \kappa) \right) \right) \right).$$
$$(17)$$

*Proof.* Suppose an input $\boldsymbol{x}$ whose prediction $h(\boldsymbol{x})$ is consistent, *i.e.*, $\exists k.\ \mu_k(\boldsymbol{x}) \ge 1 - \kappa$. Let $k^*$ denote this predicted class. Let $\phi_1$ and $\phi_2$ denote two scalar distances to shift $\boldsymbol{x}$ and assume the following features of these two distances. First, $\Delta_{\max}\mu_{k^*}(\phi_1) = 1/2 - \kappa$ and $\Delta\mu_{k^*}(\phi_2, \hat{\boldsymbol{\phi}}_2) > 1/2 - \kappa$. The latter indicates that in some direction, moving the input by a distance of $\phi_2$ results in a change in $\mu_{k^*}$ greater than $1/2 - \kappa$. Second, $\phi_1 > \phi_2$. From the first condition, we can derive that

$$\min_{\hat{\boldsymbol{\phi}}_1 \in \mathbb{S}^{n-1}} \int_{\boldsymbol{t} \in \mathbb{R}^n} \min\left( v(\boldsymbol{t} - \phi_1\hat{\boldsymbol{\phi}}_1), v(\boldsymbol{t}) \right) d\boldsymbol{t} > \int_{\boldsymbol{t} \in \mathbb{R}^n} \min\left( v(\boldsymbol{t} - \phi_2\hat{\boldsymbol{\phi}}_2), v(\boldsymbol{t}) \right) d\boldsymbol{t}. \qquad (18)$$

In other words, we can find some vector $\phi_2 \hat{\boldsymbol{\phi}}_2$ such that this shift results in a vicinity overlap smaller than the minimum vicinity overlap caused by a $\phi_1$-magnitude shift.

Next, we further shift $\boldsymbol{x} + \phi_2 \hat{\boldsymbol{\phi}}_2$ along $\hat{\boldsymbol{\phi}}_2$ direction but with the magnitude $\phi_1 - \phi_2$. This new position, $\boldsymbol{x} + \hat{\boldsymbol{\phi}}_2 \phi_1$, is farther away from $\boldsymbol{x}$ than $\boldsymbol{x} + \phi_2 \hat{\boldsymbol{\phi}}_2$ is because $\phi_1 > \phi_2$. Additionally, $\boldsymbol{x} + \hat{\boldsymbol{\phi}}_2 \phi_1$ results in at most the same vicinity overlap (size) as $\boldsymbol{x} + \phi_2 \hat{\boldsymbol{\phi}}_2$ does because $v$ is quasiconcave. Formally,

$$\int_{\boldsymbol{t} \in \mathbb{R}^n} \min\left(v(\boldsymbol{t} - \phi_2 \hat{\boldsymbol{\phi}}_2), v(\boldsymbol{t})\right) d\boldsymbol{t} \geq \int_{\boldsymbol{t} \in \mathbb{R}^n} \min\left(v(\boldsymbol{t} - \phi_1 \hat{\boldsymbol{\phi}}_2), v(\boldsymbol{t})\right) d\boldsymbol{t}. \tag{19}$$

Observe that Inequality (19) contradicts (18). Therefore, the two assumptions cannot hold simultaneously. An adversarial example requires $\boldsymbol{x}' \in \mathbb{V}(\boldsymbol{x}), \mu_{k^*}(\boldsymbol{x}') \leq 1/2$. Thus, if $\Delta_{\max} \mu_{k^*}(\phi_1) = 1/2 - \kappa$, the distance between a consistent input and any of its adversarial samples is greater than (or equal to) $\phi_1$. The rationale of $1/2 - \kappa$ is proven in Appendix B.5. $\qquad \square$

Intuitively, at optimal probabilistic robust accuracy, if an input has a probabilistically consistent prediction, all its neighbours in some specific vicinity are predicted the same. Namely, the prediction at this input is deterministically robust within this (likely smaller) vicinity. Corollary 3.5 suggests that the bounds stated in Theorem 3.4 persist even as the shift approaches zero.

**Corollary 3.5.** *There exists a finite real value such that for all inputs, the directional derivative value with respect to any arbitrary nonzero vector $\hat{\boldsymbol{\phi}}$ (unit vector) does not exceed this finite value and does not fall below the negative of this value. Formally: where $\cdot$ denotes the dot product,*

$$\exists b \in \mathbb{R}, \forall \boldsymbol{x} \in \mathbb{R}^n, \forall \hat{\boldsymbol{\phi}} \in \mathbb{S}^{n-1}, \; -b \leq \nabla \mu_k(\boldsymbol{x}) \cdot \hat{\boldsymbol{\phi}} \leq b. \tag{20}$$

Proof of Corollary 3.5 is provided in Appendix B.6. A one-dimensional example in Appendix B.7 illustrates this idea. We then present an implication of Theorem 3.4 for $L^\infty$-norm in Corollary 3.6.

**Corollary 3.6.** *If $h^*$ is optimal for the probabilistic robustness with respect to an $L^\infty$-vicinity on distribution $D$, then the vicinity size of the deterministically robust vicinity $\mathbb{V}^{\downarrow \kappa}$ around each probabilistically consistent input is $\epsilon(1 - (2\kappa)^{\frac{1}{n}})$.*

Proof of Corollary 3.6 is in Appendix B.8. We visualise its effect using a two-dimensional example in Figure 1d. Let $\boldsymbol{x}, \boldsymbol{x}'$ be two probabilistically consistent inputs and $h(\boldsymbol{x}) \neq h(\boldsymbol{x}')$. To minimise $|\boldsymbol{x}' - \boldsymbol{x}|$, $\boldsymbol{x}'$ must be in the diagonal direction (as in Corollary 3.6). The shift from $\boldsymbol{x}$ to $\boldsymbol{x}'$ is $-2(\phi_1 \hat{\boldsymbol{x}_1} + \phi_2 \hat{\boldsymbol{x}_2})$. Each triangle accounts for $\kappa$ of the original vicinity size (volume). Box $\mathbb{V}^{\downarrow \kappa}$ has side length $2\phi_i$. Thus, solving $(2\epsilon - 2\phi_i)^2 = 2\kappa(2\epsilon)^2$, we get $\mathbb{V}^{\downarrow \kappa}$ has vicinity size $\phi_1 = \phi_2 = (1 - \sqrt{2\kappa})\epsilon$. Although Corollary 3.6 concerns $L^\infty$, we can analyse other types similarly, *i.e.*, find the direction with the fastest vicinity overlap decrease and measure the distance to the nearest adversarial example.

In brief, an optimal probabilistic robust accuracy has the following implications. Theorem 3.4 suggests that each probabilistic robust input is also deterministically robust, but within a much smaller vicinity. Corollary 3.6 further quantifies the size of this vicinity. Particularly, the order $1/n$ lets vicinity shrink fast as $n$ grows, making probabilistic robust accuracy higher. Overall, deterministic robust accuracy with the smaller vicinity bounds probabilistic robust accuracy. This is formally captured in Theorem 3.7. The effect of this reduced Bayes uncertainty is illustrated in Figure 1c.

**Theorem 3.7.** *Given distribution $D$, vicinity with size $\epsilon$, and tolerance level $\kappa$, the probabilistic robust accuracy has an upper bound as shown in Equation (21), where $\mathbb{V}^{\downarrow \kappa}$ or $v^{\downarrow \kappa}$ is the "smaller" vicinity and $\mathbb{K}$ follows the definition in Equation (7), denoting the domain near the boundary.*

$$\Upsilon_{\text{prob}}^+ (D, h, \mathcal{V}, \kappa) \leq \max_{h \in \{\mathbb{R}^n \rightarrow \{0, 1, \ldots, K-1\}\}} \Upsilon_{\text{rob}}^+ \left(D, h, \mathbb{V}^{\downarrow \kappa}\right)$$
$$= \mathrm{E}_{(\mathbf{x}, \mathbf{y}) \sim (D * v^{\downarrow \kappa})} \left[\max_k p(\mathbf{y} = k | \mathbf{x}) \mathbf{1}_{\mathbf{x} \notin \mathbb{K}_{\lceil D \rceil * v^{\downarrow \kappa}}}\right] \tag{21}$$

*Proof.* Theorem 3.4 infers $\forall \boldsymbol{x}, \left((P_{\mathbf{x}' \sim \mathcal{V}(\boldsymbol{x})}(h(\mathbf{x}') \neq h(\boldsymbol{x})) \leq \kappa\right) \rightarrow \forall \boldsymbol{x}' \in \mathbb{V}^{\downarrow \kappa}(\boldsymbol{x}). \; h(\boldsymbol{x}') = h(\boldsymbol{x}))$ at optimal probabilistic robust accuracy. Consider this expression in the form of $\forall \boldsymbol{x}, \text{Event}_1(\boldsymbol{x}) \rightarrow \text{Event}_2(\boldsymbol{x})$. This implies that the occurrence rate of Event 2 (deterministic robust accuracy with $\mathbb{V}^{\downarrow \kappa}$) always upper bounds the rate of Event 1 (probabilistic robust accuracy with $\mathbb{V}$). Additionally, the

deterministic robust accuracy with $\mathbb{V}^{\downarrow \kappa}$ has an upper bound, as what has been shown in (Zhang & Sun, 2024). $\qquad\square$

Besides the upper bound, we also find a relatively loose lower bound of probabilistic robust accuracy as shown in Corollary 3.8. This lower bound is the deterministic robust accuracy when the vicinity size is the same as the vicinity size assumed for probabilistic robustness.

**Corollary 3.8.** *The upper bound of probabilistic robust accuracy monotonically increases as $\kappa$ grows. Further, for all tolerance levels $\kappa$, the upper bound of probabilistic robust accuracy lies between the upper bound of deterministic robust accuracy and the upper bound of vanilla accuracy. Formally,*

$$\forall \kappa_1, \kappa_2. \quad (\kappa_1 < \kappa_2) \rightarrow \min_h \Upsilon_{\text{prob}}^+(D, h, \mathcal{V}, \kappa_1) \leq \min_h \Upsilon_{\text{prob}}^+(D, h, \mathcal{V}, \kappa_2)$$
$$\forall \kappa. \quad \min_h \Upsilon_{\text{rob}}^+(D, h, \mathbb{V}) \leq \min_h \Upsilon_{\text{prob}}^+(D, h, \mathcal{V}, \kappa) \leq \min_h \Upsilon_{\text{acc}}^+(D, h) \tag{22}$$

*Proof.* Let $\kappa_1 < \kappa_2$ and $k^* = \arg\max_k \mu_k(\boldsymbol{x})$, and then we get the sign of $e(\boldsymbol{x}, \kappa_1) - e(\boldsymbol{x}, \kappa_2)$ is the same as $u(\kappa_2 - 1 + \mu_{k^*}(\boldsymbol{x})) - u(\kappa_1 - 1 + \mu_{k^*}(\boldsymbol{x}))$. This is because the posterior probability $P(\mathbf{y} \mid \mathbf{x} = \boldsymbol{x})$ is non-negative. Since $\kappa_1 < \kappa_2$, and unit step function monotonically increases, we get $e(\boldsymbol{x}, \kappa_1) - e(\boldsymbol{x}, \kappa_2) \geq 0$. A smaller $\kappa$ leads to a lower or equal upper bound of probabilistic robust accuracy. For deterministic robust accuracy, $\kappa = 0$, which is the least value. Thus, deterministic robust accuracy has a smaller upper bound than probabilistic robust accuracy does.

On the other hand, from the intuition of error combination in Section 3.1, we get that the combined error is $e(\boldsymbol{x}) = 1 - (1 - e_{\text{cns}}(\boldsymbol{x}, h; \mathcal{V}, \kappa))(1 - e_{\text{cor}}(\boldsymbol{x}, h; P(\mathbf{y} \mid \boldsymbol{x})))$. As $0 \leq e_{\text{cns}} \leq 1$, we get $e(\boldsymbol{x}) \geq 1 - (1 - e_{\text{cor}}(\boldsymbol{x}, h; \kappa))$. Note that the expectation of $e_{\text{cor}}(\boldsymbol{x}, h; \kappa)$ is the error in vanilla accuracy. Thus, $\Upsilon_{\text{prob}}^+(D, h, \mathcal{V}, \kappa) \leq \Upsilon_{\text{acc}}^+(D, h)$. So it is with their upper bounds. We also provide an extended proof for this theorem in Appendix B.9. $\qquad\square$

In summary, we show that probabilistic robust accuracy is both lower and upper bounded. We also show why the upper bound of probabilistic robust accuracy can be much greater than that of deterministic robust accuracy. Intuitively, our result suggests that probabilistic robustness indeed allows us to sacrifice much less accuracy, compared to that of deterministic robustness. Furthermore, adopting a larger (more relaxed) $\kappa$ (up to 1/2) can effectively increase probabilistic robust accuracy.

## 4 EXPERIMENT

We conduct experiments[1] to validate the above established results empirically. Note that Theorem 3.2 infers voting is optimal. Section 3.2 establishes the probabilistic robust accuracy upper bound. Do these match empirical results? Further, ablation experiments on real-world distribution study how this upper bound changes as $\kappa$ grows. The relationship between (vanilla) accuracy, probabilistic, and deterministic robust accuracy is also studied. In the following, we describe the setups and then answer these questions.

**Setup** Our setup follows prior Bayes error studies (Ishida et al., 2023; Zhang & Sun, 2024). We include four datasets: Moons (Pedregosa et al., 2011), Chan (Chen et al., 2023), Fashion-MNIST (Xiao et al., 2017), and CIFAR-10 (Krizhevsky et al., 2009). Given each dataset, we apply a direct method (Ishida et al., 2023) to compute the Bayes error. $L^\infty$-vicinity is set with $\epsilon = 0.15, 0.15, 0.1, 2/255$ for defining robustness on respective distribution. For deterministic robustness, the Bayes error follows Equation (7) (Zhang & Sun, 2024). For probabilistic robustness, we set $\kappa = 0.1$ by default (Robey et al., 2022) and vary $\kappa$ only for the ablation study. More details are in Appendix C.1. Statistical significance is included in Appendix C.2.

**Does voting always increase probabilistic robust accuracy empirically?** As stated in Theorem 3.2, we first estimate the probabilistic robust accuracy of some classifier $h$. We then estimate that of a voting classifier $h^\dagger$, where $h^\dagger(\boldsymbol{x}) = \arg\max_k P_{\mathbf{t} \sim \mathcal{V}(\boldsymbol{x})}(k = h(\mathbf{t}))$, with sample size 100. They are compared in Table 1. Training algorithms of $h$ include data augmentation (DA, Shorten &

---

[1]Available at `https://github.com/soumission-anonyme/irreducible.git`

Table 1: Probabilistic robustness of classifiers before and after voting. $\kappa = 0.1$ and $\epsilon = 0.15, 0.15, 0.1, 2/255$ for Moons, Chan, FashionMNIST, and CIFAR-10.

| | Moons | | Chan | | FashionMNIST | | CIFAR-10 | |
|---|---|---|---|---|---|---|---|---|
| DA (Shorten & Khoshgoftaar, 2019) | 85.35 | **85.60** (+0.3%) | 67.96 | **68.86** (+0.9%) | 84.12 | **87.48** (+3.7%) | 76.07 | **81.38** (+5.3%) |
| RS (Cohen et al., 2019) | 84.76 | **85.18** (+0.4%) | 64.67 | **66.77** (+1.9%) | 86.29 | **88.13** (+2.1%) | 87.98 | **88.95** (+1.0%) |
| CVaR (Robey et al., 2022) | 85.52 | **85.66** (+0.1%) | 69.46 | **70.05** (+0.6%) | 88.50 | **91.07** (+2.6%) | 90.63 | **90.77** (+0.1%) |

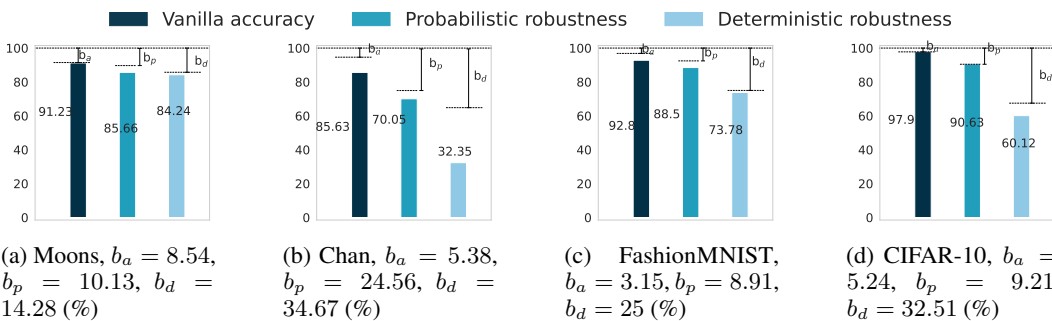

(a) Moons, $b_a = 8.54$, $b_p = 10.13$, $b_d = 14.28$ (%)

(b) Chan, $b_a = 5.38$, $b_p = 24.56$, $b_d = 34.67$ (%)

(c) FashionMNIST, $b_a = 3.15, b_p = 8.91$, $b_d = 25$ (%)

(d) CIFAR-10, $b_a = 5.24$, $b_p = 9.21$, $b_d = 32.51$ (%)

Figure 2: Comparing the SOTA classifier performance with upper bounds, *i.e.*, 1 - the Bayes error of vanilla accuracy ($b_a$), probabilistic robust accuracy ($b_p$), and deterministic robust accuracy ($b_d$).

Khoshgoftaar, 2019), randomised smoothing (RS, Cohen et al., 2019), and condition value-at-risk (CVaR, Robey et al., 2022) which is state-of-the-art (SOTA) for probabilistic robust accuracy. Note that voting always improves probabilistic robust accuracy (at least by +0.1% or on average +1.58% ). On DA and RS, the increase is significant (avg + 1.95%), partly because they are not designed specifically for probabilistic robustness. On CVaR, while modest (avg + 0.85%), we do observe an increase. This trend is maintained with a larger voting sample size (Appendix C.3).

**Is our upper bound empirically valid on existing neural networks?** To check if indeed all trained classifiers respect the theoretical upper bound of probabilistic robust accuracy on any distribution, we compare the SOTA CVaR training and the bound. The middle column of each plot in Figure 2 demonstrates this comparison. We observe that the SOTA probabilistic robust accuracy never exceeds our theoretical bound. Intriguingly, on certain distributions like CIFAR-10, SOTA training almost meets its upper bound with a small gap (0.2%), while on others, a gap remains (on average 4.06%). Theoretically, a negative gap may also occur when a classifier overfits the data samples (Ishida et al., 2023). Our upper bound is empirically useful in approximating the room for improvement.

**How does probabilistic robust accuracy compare to vanilla accuracy and deterministic robust accuracy in terms of upper bounds?** We observe in Figure 2 that invariably, the upper bound of probabilistic robust accuracy is lower than that of vanilla accuracy and higher than that of deterministic robust accuracy. In high-dimensional distributions, the upper bound of probabilistic robust accuracy is close to that of vanilla accuracy but over 27% higher than that of deterministic robust accuracy. This could be a result of the curse of dimensionality and much-reduced vicinity size according to Corollary 3.6. On Chan, the upper bound of probabilistic robustness is close to that of deterministic robust accuracy but over 20% lower than that of vanilla accuracy. This could be due to the high-frequency features in the distribution (Zhang & Sun, 2024). On Moons, these three bounds are close (at most 7% difference). The reason could be that this distribution is relatively smooth.

**What is the effect of $\kappa$ on the upper bound of probabilistic robust accuracy?** Given different $\kappa$, the upper bound of probabilistic robust accuracy can be different. We vary $\kappa$ in $[0, 0.5)$ increasing each time by 0.01. Figure 3 shows that this upper bound monotonically grows as $\kappa$ grows, which matches Corollary 3.8. Besides, the growth is fast when $\kappa$ is small (slope $> 3$ at $\kappa = 0.1$), and the growth rate decreases as $\kappa$ grows (slope $< 0.02$ at $\kappa = 0.4$). Especially, for high-dimensional distributions, a small change in $\kappa$ when $\kappa$ is small, *e.g.* $< 0.1$ can significantly increase the upper bound. This could be explained by the $1/n$ order in Corollary 3.6. This is encouraging as it shows that by sacrificing deterministic robustness only slightly, we can already improve the accuracy significantly.

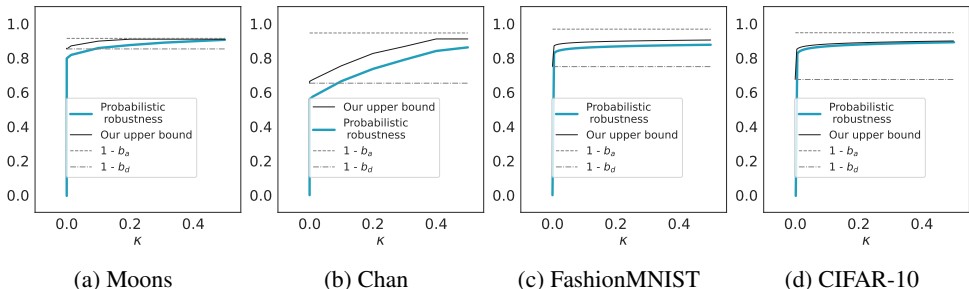

(a) Moons      (b) Chan      (c) FashionMNIST      (d) CIFAR-10

Figure 3: As $\kappa$ increases, we plot the upper bounds of probabilistic robust accuracy as well as classifiers' probabilistic robust accuracy change.

## 5 RELATED WORK

This work is closely related to studies on Bayes errors and probabilistic robustness. Computing the Bayes error of a given distribution has been studied for over half a century (Fukunaga & Hostetler, 1975), and one interesting topic is to derive or empirically estimate the upper and lower bounds of the Bayes error. Various $f$-divergences, such as the Bhattacharyya distance (Fukunaga, 1990) or the Henze-Penrose divergence (Berisha et al., 2016; Sekeh et al., 2020), have been studied. Other approaches include directly estimating the Bayes error with $f$-divergence representation instead of using a bound (Noshad et al., 2019), computing the Bayes error of generative models learned using normalizing flows (Kingma & Dhariwal, 2018; Theisen et al., 2021), or evaluate the Bayes error from data sample reassessment (Ishida et al., 2023). Recent studies apply Bayes error estimation to deterministic robustness beyond a vanilla accuracy perspective (Zhang & Sun, 2024). Our study extends this line of research and focuses on probabilistic robustness.

Improving robustness is a core topic in the recent decade (Zhang et al., 2023a; Wang et al., 2021). Adversarial training considers adversarial samples in the training phase (Madry et al., 2018; Goodfellow et al., 2015; Zhang et al., 2019a; Wang et al., 2020). However, adversarially trained models do not come with a theoretical guarantee (Zhang et al., 2018; Singh et al., 2019; Balunovic & Vechev, 2020). Certified training provides this guarantee by optimising bounds from formal verification during training but compromises performance on clean inputs (Shi et al., 2021; Müller et al., 2022). To mitigate these problems, probabilistic robustness methods such as PRoA (Zhang et al., 2023b) or CVaR (Robey et al., 2022) are proposed. Probabilistic robustness offers a desirable balance between robustness and accuracy, making it more applicable in real-world scenarios.

## 6 CONCLUSION

We investigate the open problem of whether there is an upper bound on probabilistic robustness. We find that the optimal prediction should be the Maximum A Posteriori of predictions in the vicinity. Then, we show that any probabilistically robust input is also deterministically robust within a smaller vicinity. As a result, the upper bound of probabilistic robust accuracy can be obtained from that of deterministic robust accuracy. We empirically validate our upper bound established through a theoretical study by comparing it with SOTA training for probabilistic robustness and the upper bounds of vanilla accuracy or deterministic robust accuracy. The experimental results match our theorems and show that our bounds could indicate room for improvement in practice. Finally, this study presents a theoretical argument supporting probabilistic robustness as the appropriate target for achieving neural network robustness.

**Limitation** A limitation of our upper bound is that Theorem 3.4 requires the posterior probability to calculate the probabilistic robust accuracy upper bound while it might be difficult to obtain the posterior for some cases. This generally occurs for Bayes uncertainty analyses (Fukunaga & Hostetler, 1975) in finding bounds of accuracy (Theisen et al., 2021; Nielsen, 2014; Moon et al., 2015) or deterministic robustness (Zhang & Sun, 2024). Yet, this can be compensated by various density estimation methods (Renggli et al., 2021; Zhang & Sun, 2024) or re-evaluating the probability from a dataset (Ishida et al., 2023).

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

## A  Notations

### A.1  Vicinity Notations

**Definition A.1** (Vicinity). Inputs within the vicinity of an input $x$ are imperceptible from $x$. We call this vicinity an $x$-vicinity. To capture imperceptibility, an $x$-vicinity can be denoted in (at least four) different but equivalent notations, *i.e.*, distance-threshold, set, distribution, and function notations. Occasionally, an input within $x$-vicinity is called a neighbour of $x$.

**Distance-threshold Notation of Vicinity**  The distance-threshold notation is one of the earliest notations to depict vicinity (Ma et al., 2018; Athalye et al., 2018; Bhattacharya & Gupta, 2019). Here, the neighbour of a sample $x \in \mathbb{X}$ refers to an input that lies within a certain threshold distance from $x$. Formally, $x'$ is within $x$-vicinity if and only if

$$d(x', x) \leq \epsilon \tag{23}$$

where $d$ measures the distance between two inputs, and this distance needs to be smaller than a threshold $\epsilon$ to be considered imperceptible.

Specifically, the distance function can be defined in a variety of ways *e.g.*, $L^p$ norm ($p = 0, 1, 2,$ or $\infty$) or domain-specific transformations that preserve labels, such as tilting or zooming.

$$d(x', x) = \|x' - x\|_p, \quad \text{(Additive in } L^p \text{ norm), or}$$

$$d(x', x) = \begin{cases} |\epsilon|, & \text{if } f_{\text{transform}}(x, \epsilon) = x', \\ \epsilon + 1, & \text{otherwise} \end{cases} \tag{24}$$

where the transformation function $f_{\text{transform}} : \mathbb{X} \to \mathbb{X}$ can be, for example, an image rotation with a parameter determining the degree of rotation.

**Set Notation of Vicinity**  Here, the vicinity of a sample $x \in \mathbb{X}$ refers to a set containing all neighbours of $x$. Given an input, all inputs whose distance to the given input is within a certain threshold form a set, defined as a vicinity of the given input. For any $x \in \mathbb{X}$, its vicinity is expressed as

$$\mathbb{V}(x) = \{ x' \mid d(x, x') \leq \epsilon \} \tag{25}$$

Since the corresponding distance function $d$ can be a representation of different distance measures, the set notation $\mathbb{V}(\boldsymbol{x})$ can also be a representation of various sets, *i.e.*, $\mathbb{V}_1(\boldsymbol{x}), \mathbb{V}_2(\boldsymbol{x})$ could be $x$-vicinities defined in two different ways.

**Function Notation of Vicinity**   The set or distance representation may be inconvenient sometimes (Zhang & Sun, 2024). We may sometimes need the notion of $\mathbf{1}_{\boldsymbol{x}' \in \mathbb{V}(\boldsymbol{x})}$ to quantify if $\boldsymbol{x}'$ is a neighbour of $\boldsymbol{x}$. For instance, if we would like to sum the marginal probability of all neighbours of $\boldsymbol{x}$, we can $\int_{\mathbb{X}} \mathbf{1}_{\boldsymbol{x}' \in \mathbb{V}(\boldsymbol{x})} p(\mathbf{x} = \boldsymbol{x}') d\boldsymbol{x}'$ instead of $\int_{\mathbb{V}(\boldsymbol{x})} p(\mathbf{x} = \boldsymbol{x}') d\boldsymbol{x}'$ to avoid a varying interval of integration.

In this case, a vicinity function, which is an equivalent form of the set, can be defined as

$$v_{\boldsymbol{x}}(\boldsymbol{x}') = \begin{cases} \left(\int_{\mathbb{V}(\boldsymbol{x})} d\boldsymbol{x}''\right)^{-1}, & \text{if } \boldsymbol{x}' \in \mathbb{V}(\boldsymbol{x}) \\ 0, & \text{otherwise} \end{cases} \tag{26}$$

Essentially, Equation (26) can be viewed as a probability density function (PDF) uniformly defined over the vicinity around an input $\boldsymbol{x}$. Now we shift the x-coordinate by $\boldsymbol{x}$, we get

$$v_{\boldsymbol{0}}(\boldsymbol{x}' - \boldsymbol{x}) = \begin{cases} \left(\int_{\mathbb{V}(\boldsymbol{0})} d\boldsymbol{x}''\right)^{-1}, & \text{if } \boldsymbol{x}' - \boldsymbol{x} \in \mathbb{V}(\boldsymbol{0}) \\ 0, & \text{otherwise} \end{cases} \tag{27}$$

Assuming that the vicinity function is translation invariant, we can drop the subscript $\boldsymbol{0}$, and use a positive constant $\epsilon_{\mathrm{v}}$ to represent $\int_{\mathbb{V}(\boldsymbol{0})} d\boldsymbol{x}''$, *i.e.*, the size of the vicinity. Thus, the vicinity function $v : \mathbb{X} \to \left\{ 0, \epsilon_{\mathrm{v}}^{-1} \right\}$ can be expressed as

$$v(\boldsymbol{x}) = \begin{cases} \epsilon_{\mathrm{v}}^{-1} & \text{if } \boldsymbol{x} \in \mathbb{V}(\boldsymbol{0}) \\ 0, & \text{otherwise} \end{cases} \tag{28}$$

An example of a one-dimensional input's vicinity is shown in Figure 4.

**Distribution Notation of Vicinity**   As outlined in Remark 2.1, when we need to sample from the vicinity, we need its distribution notation. The distribution notation for $\boldsymbol{x}$-vicinity is $\mathcal{V}(\boldsymbol{x})$ whose PDF is denoted as $v : \mathbb{R}^n \to \mathbb{R}$. If $v(\boldsymbol{x}' - \boldsymbol{x}) > 0$, we say $\boldsymbol{x}'$ is within $\boldsymbol{x}$-vicinity.

An imperceptible perturbation from any $\boldsymbol{x}$ to $\boldsymbol{x}'$ means that $\boldsymbol{x}'$ is a 'neighbour' of $\boldsymbol{x}$, or equivalently, $\boldsymbol{x}'$ is in the $\boldsymbol{x}$-vicinity. $\boldsymbol{x}$-vicinity is a (probabilistic) distribution $\mathcal{V}(\boldsymbol{x})$ centred at $\boldsymbol{x}$. A standard vicinity $\mathcal{V}(\boldsymbol{0})$ is centred at the origin and its PDF is denoted as $v : \mathbb{R}^n \to \mathbb{R}$. Thus, the PDF centred any specific $\boldsymbol{x}$ would be $v(\boldsymbol{x}' - \boldsymbol{x})$.

$v$ is typically an even and quasiconcave function. Formally,

$$\begin{aligned} &v(\boldsymbol{x}) = v(-\boldsymbol{x}) \\ &\quad \forall t \in [0, 1] \, \forall \boldsymbol{x_1}, \boldsymbol{x_2} \in \mathbb{R}^n, \quad v\left(t\boldsymbol{x_1} + (1 - t)\boldsymbol{x_2}\right) \geq \min\left(v(\boldsymbol{x_1}), v(\boldsymbol{x_2})\right) \end{aligned} \tag{29}$$

A uniform $L^p$ $\boldsymbol{x}$-vicinity assumes that all inputs outside an $L^p$-norm of $\boldsymbol{x}$ are distinguishable from $\boldsymbol{x}$ and all inputs within this norm are equally imperceptible from $\boldsymbol{x}$. This $L^p$ vicinity function is captured in Equation (30) where parameter $\epsilon$ specifies a size.

$$v(\boldsymbol{x}' - \boldsymbol{x}) = \begin{cases} \frac{\Gamma(1 + n/p)}{(2\epsilon \Gamma(1 + 1/p))^n}, & \text{if } \|\boldsymbol{x}' - \boldsymbol{x}\|_p \leq \epsilon \\ 0, & \text{otherwise} \end{cases} \tag{30}$$

The fraction in Equation (30) represents the inverse of the $L^p$-norm volume, where $\Gamma$ denotes the gamma function. Vicinity functions assess the likelihood of $\boldsymbol{x}'$ being a neighbour to $\boldsymbol{x}$. In the uniform $L^p$-norm context, all inputs within the norm are equally valid as neighbours and no inputs outside the norm are neighbours.

*Remark* A.2. Adversarial samples of an input $\boldsymbol{x}$ always reside in $\boldsymbol{x}$-vicinity.

## B   COMPLETE PROOFS AND DERIVATIONS

This section provides detailed proofs for various lemmas, theorems, or corollaries. For each, we restate the original claim followed by a more comprehensive proof than what appears in the main text. Additionally, we include detailed derivations for certain equations not covered in the theorems.

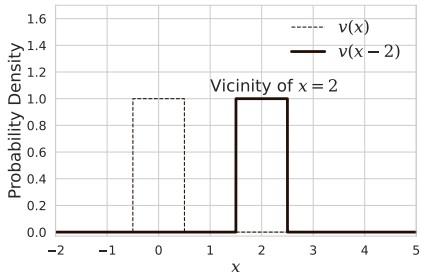

Figure 4: 1D visualizations of vicinity function as outlined in Remark 2.1. This vicinity function is a rectangular function that returns a constant value if an input is in the vicinity. Vicinity function $v(\boldsymbol{x})$ is shown in dashed line ($\epsilon = 0.5$). To get the vicinity at a specific input $\boldsymbol{x} = 2$, we simply shift $v(\boldsymbol{x})$ along the positive direction of the x-axis by 2.

### B.1 PROOF OF LEMMA 3.1

**Lemma 3.1.** *For the predictions within the vicinity (of an input $\boldsymbol{x}$) to be consistent, at most one class has a prediction probability exceeding $1 - \kappa$ in this vicinity. Thus, $\kappa < \frac{1}{2}$.*

*Proof.* Assume that $\kappa > \frac{1}{2}$ such that any $\sum_{k=0}^{K-1} \mu_k(\boldsymbol{x}) \mathbf{1}_{h(\boldsymbol{x})=k}$ that is greater than or equal to $\frac{1}{2}$ satisfies the consistency condition because $\frac{1}{2} > 1 - \kappa$. Hence, there may exist some $k_1 \neq k_2$ such that $\mu_{k_1} + \mu_{k_2} \leq 1$ and

$$\sum_{k=0}^{K-1} \mu_{k_1}(\boldsymbol{x}) \mathbf{1}_{h(\boldsymbol{x})=k_1} = \sum_{k=0}^{K-1} \mu_{k_2}(\boldsymbol{x}) \mathbf{1}_{h(\boldsymbol{x})=k_2} > 1 - \kappa. \tag{31}$$

The existence of such distinct indices $k_1, k_2$ implies that if prediction at $\boldsymbol{x}$ is $k_1$, it is consistent with its neighbours. Similarly, if prediction at $\boldsymbol{x}$ is $k_2$, it is consistent with its neighbours. However, since $k_1 \neq k_2$, it is not possible for the neighbours' predictions to simultaneously be consistent with both $k_1$ and $k_2$. This scenario contradicts Inequality (31), and thus the initial assumption does not hold. $\square$

### B.2 DERIVATION OF THE COMBINED ERROR CONSIDERING PROBABILISTIC ROBUSTNESS

Considering probabilistic robustness, the error at input $\boldsymbol{x}$ is a combined error of $e_{\text{cor}}$ and $e_{\text{cns}}$ at $\boldsymbol{x}$. As discussed, we use two intuitions to derive the combined error. First, if $e_{\text{cns}}(\boldsymbol{x}, h; \kappa) = 1$, the combined error is always 1. Second, if $e_{\text{cns}}(\boldsymbol{x}, h; \mathcal{V}, \kappa) = 0$, the combined error equals $e_{\text{cor}}(\boldsymbol{x}, h; P(\text{y} \mid \boldsymbol{x}))$. Note that inconsistency is a binary value that takes either 0 or 1 while incorrectness takes a real value from 0 to 1 depending on the posterior at input. In the following, we derive the combined error $e(\boldsymbol{x}, h; P(\text{y} \mid \boldsymbol{x}), \kappa)$ as expressed in Equation (12).

$$
\begin{aligned}
e(\boldsymbol{x}, h; P(\text{y}|\boldsymbol{x}), \kappa) &= (1 - e_{\text{cns}}(\boldsymbol{x}, h; \mathcal{V}, \kappa)) e_{\text{cor}}(\boldsymbol{x}, h; P(\text{y} \mid \boldsymbol{x})) + e_{\text{cns}}(\boldsymbol{x}, h; \mathcal{V}, \kappa) \\
&= e_{\text{cor}}(\boldsymbol{x}, h; P(\text{y} \mid \boldsymbol{x})) - e_{\text{cor}}(\boldsymbol{x}, h; P(\text{y} \mid \boldsymbol{x})) e_{\text{cns}}(\boldsymbol{x}, h; \mathcal{V}, \kappa) + e_{\text{cns}}(\boldsymbol{x}, h; \mathcal{V}, \kappa) \\
&= e_{\text{cor}}(\boldsymbol{x}, h; P(\text{y} \mid \boldsymbol{x})) + e_{\text{cns}}(\boldsymbol{x}, h; \mathcal{V}, \kappa) - e_{\text{cor}}(\boldsymbol{x}, h; P(\text{y} \mid \boldsymbol{x})) e_{\text{cns}}(\boldsymbol{x}, h; \mathcal{V}, \kappa) \\
&= 1 - (1 - e_{\text{cns}}(\boldsymbol{x}, h; \mathcal{V}, \kappa))(1 - e_{\text{cor}}(\boldsymbol{x}, h; P(\text{y} \mid \boldsymbol{x}))) \\
&= 1 - \left(1 - u\left(1 - \kappa - \sum_{k=0}^{K-1} \mu_k(\boldsymbol{x}) \mathbf{1}_{h(\boldsymbol{x})=k}\right)\right) \left(\sum_{y=0}^{K-1} P(\text{y} = y \mid \mathbf{x} = \boldsymbol{x}) \mathbf{1}_{h(\boldsymbol{x})=y}\right) \\
&= 1 - u\left(\kappa - 1 + \sum_{k=0}^{K-1} \mu_k(\boldsymbol{x}) \mathbf{1}_{h(\boldsymbol{x})=k}\right) \left(\sum_{y=0}^{K-1} P(\text{y} = y \mid \mathbf{x} = \boldsymbol{x}) \mathbf{1}_{h(\boldsymbol{x})=y}\right)
\end{aligned}
\tag{32}
$$

## B.3 EXTENDED PROOF OF THEOREM 3.2

**Theorem 3.2.** *If $h^*$ is optimal for the probabilistic robustness on a given distribution,* i.e., $h^* = \arg\min_h \int_{\boldsymbol{x}\in\mathbb{R}^n} e(\boldsymbol{x})p(\mathbf{x}=\boldsymbol{x})d\boldsymbol{x}$, *we would always have* $\forall \boldsymbol{x}\in\mathbb{R}^n, h^*(\boldsymbol{x}) = \arg\max_k \mu_k(\boldsymbol{x})$.

*Proof.* Let $h_1$ and $h_2$ be two distinct classification functions such that $h_1(\boldsymbol{x}) = \arg\max_k \mu_k(\boldsymbol{x})$ and $h_2(\boldsymbol{x}) \neq h_1(\boldsymbol{x})$. We want to prove $e(\boldsymbol{x}, h_1) \leq e(\boldsymbol{x}, h_2)$, such that $h_1$ must be optimal for probabilistic robustness. First, we denote $k_1, k_2 \in \{0, 1, ..., K-1\}, k_1 = h_1(\boldsymbol{x}), k_2 = h_2(\boldsymbol{x}) \neq k_1$. Then,

$$
\begin{aligned}
e(\boldsymbol{x}, h_1) - e(\boldsymbol{x}, h_2) &= 1 - u\left(\kappa - 1 + \sum_{k=0}^{K-1} \mu_k(\boldsymbol{x})\, \mathbf{1}_{h_1(\boldsymbol{x})=k}\right)\left(\sum_{y=0}^{K-1} P(\mathbf{y}=y|\mathbf{x}=\boldsymbol{x})\, \mathbf{1}_{h_1(\boldsymbol{x})=y}\right) \\
&\quad - 1 + u\left(\kappa - 1 + \sum_{k=0}^{K-1} \mu_k(\boldsymbol{x})\, \mathbf{1}_{h_2(\boldsymbol{x})=k}\right)\left(\sum_{y=0}^{K-1} P(\mathbf{y}=y|\mathbf{x}=\boldsymbol{x})\, \mathbf{1}_{h_2(\boldsymbol{x})=y}\right) \\
&= u\big(\kappa - 1 + \mu_{k_2}(\boldsymbol{x})\big)P(\mathbf{y}=k_2 \mid \mathbf{x}=\boldsymbol{x}) - u\left(\kappa - 1 + \mu_{k_1}(\boldsymbol{x})\right)P(\mathbf{y}=k_1 \mid \mathbf{x}=\boldsymbol{x}).
\end{aligned}
\tag{33}
$$

Since $\mu_{k_2}(\boldsymbol{x}) \leq \mu_{k_1}(\boldsymbol{x})$, we get $\mu_{k_2}(\boldsymbol{x}) \leq 1/2$. Recall $\kappa < 1/2$ from Lemma 3.1, we get

$$
\kappa - 1 + \mu_{k_2}(\boldsymbol{x}) \; < \; \frac{1}{2} - 1 + \frac{1}{2} = 0
\tag{34}
$$

Consequently, when the input of a unit step function is negative, we have $u\left(\kappa - 1 + \mu_{k_2}(\boldsymbol{x})\right) = 0$. Therefore, we get the following expression where the value of a unit step function and the value of conditional probability are both non-negative.

$$
e(\boldsymbol{x}, h_1) - e(\boldsymbol{x}, h_2) = -u\left(\kappa - 1 + \mu_{k_1}(\boldsymbol{x})\right)P(\mathbf{y}=k_1|\mathbf{x}=\boldsymbol{x}) \leq 0
\tag{35}
$$

Hence, the error of $h_1$ is no greater than the error of $h_2$. This inequality applies to any input $\boldsymbol{x}$. Hence, a classification function like $h_1$ is optimal. $\qquad\square$

## B.4 PROOF OF LEMMA 3.3

**Lemma 3.3.** *A change in $\mu_k$ results from shifting an input by a certain distance $\phi$ within the vicinity. This change is bounded in any direction $\hat{\phi}$. Formally,*

$$
\forall \boldsymbol{x} \in \mathbb{R}^n, \forall \phi \in \mathbb{R}, \left( \left(\forall \hat{\phi} \in \mathbb{S}^{n-1},\, v\left(\frac{\phi}{2}\hat{\phi}\right) > 0\right) \to \right.
$$

$$
\left. \forall \hat{\phi} \in \mathbb{S}^{n-1}, \left(\left|\mu_k(\boldsymbol{x}+\phi\hat{\phi}) - \mu_k(\boldsymbol{x})\right| \leq \left|1 - \min_{\hat{\phi}'\in\mathbb{S}^{n-1}} \int_{\boldsymbol{t}\in\mathbb{R}^n} \min\left(v(\boldsymbol{t}-\phi\hat{\phi}'), v(\boldsymbol{t})\right) d\boldsymbol{t}\right|\right) \right),
\tag{36}
$$

*where $\in \mathbb{S}^{n-1}$ denotes the set of all unit vectors in $\mathbb{R}^n$.*

*Proof.* Each $\mu_k$ has a convolutional form as provided in Equation (14). Therefore, the change of $\mu_k$ resulting from a shift with magnitude $\phi$ in direction $\phi\hat{\phi}$ can be expressed as

$$
\mu_k(\boldsymbol{x}+\phi\hat{\phi}) - \mu_k(\boldsymbol{x}) = \int_{\boldsymbol{t}\in\mathbb{R}^n} \mathbf{1}_{h(\boldsymbol{t})=k}\, v(\boldsymbol{x}+\phi\hat{\phi}-\boldsymbol{t})\, d\boldsymbol{t} - \int_{\boldsymbol{t}\in\mathbb{R}^n} \mathbf{1}_{h(\boldsymbol{t})=k}\, v(\boldsymbol{x}-\boldsymbol{t})\, d\boldsymbol{t}
\tag{37}
$$

For simplicity, let $\boldsymbol{\phi} = \phi\hat{\phi}$ for the moment. Then, according to the linearity of integration, we can put the two integrands under the same integral as

$$
\mu_k(\boldsymbol{x}+\boldsymbol{\phi}) - \mu_k(\boldsymbol{x}) = \int_{\boldsymbol{t}\in\mathbb{R}^n} \left(\mathbf{1}_{h(\boldsymbol{t})=k}\, v(\boldsymbol{x}+\boldsymbol{\phi}-\boldsymbol{t}) - \mathbf{1}_{h(\boldsymbol{t})=k}\, v(\boldsymbol{x}-\boldsymbol{t})\right) d\boldsymbol{t}
\tag{38}
$$

Observe that we can combine like terms $\mathbf{1}_{h(\boldsymbol{t})=k}$ shared by two parts of the integrands. Thus,

$$
\mu_k(\boldsymbol{x}+\boldsymbol{\phi}) - \mu_k(\boldsymbol{x}) = \int_{\boldsymbol{t}\in\mathbb{R}^n} \mathbf{1}_{h(\boldsymbol{t})=k}\left(v(\boldsymbol{x}+\boldsymbol{\phi}-\boldsymbol{t}) - v(\boldsymbol{x}-\boldsymbol{t})\right) d\boldsymbol{t}
\tag{39}
$$

Applying the symmetry of the vicinity function about axes, we can get

$$\mu_k(\boldsymbol{x} + \boldsymbol{\phi}) - \mu_k(\boldsymbol{x}) = \int_{\boldsymbol{t} \in \mathbb{R}^n} \mathbf{1}_{h(\boldsymbol{t})=k} \left( v(\boldsymbol{t} - \boldsymbol{x} - \boldsymbol{\phi}) - v(\boldsymbol{t} - \boldsymbol{x}) \right) d\boldsymbol{t} \tag{40}$$

Next, we shift the integral limit by $+\boldsymbol{x}$, the integrand becomes $\mathbf{1}_{h(\boldsymbol{t}+\boldsymbol{x})=k}(v(\boldsymbol{t} - \boldsymbol{\phi}) - v(\boldsymbol{t}))$, and the interval of integration remains the same. To find the upper and lower bounds of $\mu_k(\boldsymbol{x} + \boldsymbol{\phi}) - \mu_k(\boldsymbol{x})$, we want to find those for this integrand. Observe that $\mathbf{1}_{h(\boldsymbol{t}+\boldsymbol{x})=k}$ either takes value 0 or 1. Thus, letting $\mathbf{1}_{h(\boldsymbol{t}+\boldsymbol{x})=k} = 1$ if and only if $(v(\boldsymbol{t} - \boldsymbol{\phi}) - v(\boldsymbol{t})) > 0$ will maximise the integrand, and letting $\mathbf{1}_{h(\boldsymbol{t}+\boldsymbol{x})=k} = 1$ if and only if $(v(\boldsymbol{t} - \boldsymbol{\phi}) - v(\boldsymbol{t})) < 0$ will minimise the integrand. Formally,

$$\min\left(0, v(\boldsymbol{t} - \boldsymbol{\phi}) - v(\boldsymbol{t})\right) \leq \mathbf{1}_{h(\boldsymbol{t}+\boldsymbol{x})=k}(v(\boldsymbol{t} - \boldsymbol{\phi}) - v(\boldsymbol{t})) \leq \max\left(0, v(\boldsymbol{t} - \boldsymbol{\phi}) - v(\boldsymbol{t})\right) \tag{41}$$

Substitute this inequality back into the integral gives rise to

$$|\mu_k(\boldsymbol{x} + \boldsymbol{\phi}) - \mu_k(\boldsymbol{x})| \leq \int_{\boldsymbol{t} \in \mathbb{R}^n} \max\left(0, v(\boldsymbol{t} - \boldsymbol{\phi}) - v(\boldsymbol{t})\right) d\boldsymbol{t} \tag{42}$$

Now we add 1 minus 1 to the right-hand side. Note that integrating a probability density function $v$ across the entire domain also equals 1.

$$
\begin{aligned}
\text{RHS} &= 1 + \int_{\boldsymbol{t} \in \mathbb{R}^n} \max\left(0, v(\boldsymbol{t} - \boldsymbol{\phi}) - v(\boldsymbol{t})\right) d\boldsymbol{t} - 1 \\
&= 1 + \int_{\boldsymbol{t} \in \mathbb{R}^n} \left(\max\left(0, v(\boldsymbol{t} - \boldsymbol{\phi}) - v(\boldsymbol{t})\right) - v(\boldsymbol{t} - \boldsymbol{\phi})\right) d\boldsymbol{t} \\
&= 1 + \int_{\boldsymbol{t} \in \mathbb{R}^n} \max\left(-v(\boldsymbol{t} - \boldsymbol{\phi}), -v(\boldsymbol{t})\right) d\boldsymbol{t} \\
&= 1 - \int_{\boldsymbol{t} \in \mathbb{R}^n} \min\left(v(\boldsymbol{t} - \boldsymbol{\phi}), v(\boldsymbol{t})\right) d\boldsymbol{t}
\end{aligned} \tag{43}
$$

Hence, the difference between $\mu_k(\boldsymbol{x} + \boldsymbol{\phi})$ and $\mu_k(\boldsymbol{x})$ is bounded by the complement of a vicinity from another vicinity shifted by $\boldsymbol{\phi}$.

Further, recall $\boldsymbol{\phi} = \phi\hat{\boldsymbol{\phi}}$, the integrand thus becomes $\min(v(\boldsymbol{t} - \phi\hat{\boldsymbol{\phi}}), v(\boldsymbol{t}))$, and

$$\min\left(v(\boldsymbol{t} - \phi\hat{\boldsymbol{\phi}}), v(\boldsymbol{t})\right) \leq \min\left(v(\boldsymbol{t}), \min_{\hat{\boldsymbol{\phi}} \in \mathbb{S}^{n-1}} v(\boldsymbol{t} - \phi\hat{\boldsymbol{\phi}})\right) \tag{44}$$

As a result, the problem of maximising the change in $\mu_k$ by a shifting magnitude $\phi$ is converted into the optimisation of finding the direction that results in the minimum overlap between the original vicinity function ($v$) and the vicinity function shifted by $\phi$ in that direction. The resulting upper bound can be expressed as Equation (16), which is re-displayed as follows.

$$\underset{\max}{\Delta} \mu_k(\phi) = \max_{\hat{\boldsymbol{\phi}} \in \mathbb{S}^{n-1}} \mu_k(\boldsymbol{x} + \phi\hat{\boldsymbol{\phi}}) - \mu_k(\boldsymbol{x}) = 1 - \min_{\hat{\boldsymbol{\phi}} \in \mathbb{S}^{n-1}} \int_{\boldsymbol{t} \in \mathbb{R}^n} \min\left(v(\boldsymbol{t} - \phi\hat{\boldsymbol{\phi}}), v(\boldsymbol{t})\right) d\boldsymbol{t} \tag{45}$$

Similarly, the lower bound is the negative of the upper bound. Also, as long as two vicinities overlap, the change of $\mu_k$ is less than 1. $\qquad\square$

## B.5 EXTENTED PROOF OF THEOREM 3.4

In the original proof of Theorem 3.4, a value $1/2 - \kappa$ is involved. Here, we explain how we get this value, and why it stands for the minimum required $\mu_k$ drop to allow an adversarial example.

*Why $1/2 - \kappa$ marks the minimum $\mu_k$ change to have an adversarial example.* For any consistent input $\boldsymbol{x}$, we have $\mu_{k^*}(\boldsymbol{x}) \geq 1 - \kappa$, where $\kappa^*$ is the major prediction in $\boldsymbol{x}$-vicinity. Consider $\boldsymbol{x}'$ as a neighbour of $\boldsymbol{x}$. If $\mu_{k^*}(\boldsymbol{x}') > 1/2$, then we know that $h(\boldsymbol{x}') = k^*$ according to Theorem 3.2, *i.e.*, $\boldsymbol{x}'$ has the same prediction as $\boldsymbol{x}$ does. In this way $\boldsymbol{x}'$ can be possibly an adversarial example of $\boldsymbol{x}$ only if $\mu_{k^*}(\boldsymbol{x}') \leq 1/2$. Thus, we get the minimum requirement of $\mu_k$ drop to allow an adversarial example to appear to be $1 - \kappa - 1/2 = 1/2 - \kappa$. $\qquad\square$

## B.6 PROOF OF COROLLARY 3.5

**Corollary 3.5.** *There exists a finite real value such that for all inputs, the directional derivative value with respect to any arbitrary nonzero vector $\hat{\phi}$ (unit vector) does not exceed this finite value and does not fall below the negative of this value. Formally, $\cdot$ denotes the dot product, and*

$$\exists b \in \mathbb{R}, \forall \boldsymbol{x} \in \mathbb{R}^n, \forall \hat{\phi} \in \mathbb{S}^{n-1}, \ -b \leq \nabla \mu_k(\boldsymbol{x}) \cdot \hat{\phi} \leq b. \tag{46}$$

*Proof.* The directional derivative of $\mu_k$ in the direction of $\hat{\phi} \in \mathbb{S}^{n-1}$ can be expressed as

$$\nabla \mu_k(\boldsymbol{x}) \cdot \hat{\phi} = \lim_{\delta \to 0} \frac{\mu_k(\boldsymbol{x} + \delta \hat{\phi}) - \mu_k(\boldsymbol{x})}{\delta} \tag{47}$$

According to Lemma 3.3, we can re-express the numerator such that the directional derivative is

$$
\begin{aligned}
\nabla \mu_k(\boldsymbol{x}) \cdot \hat{\phi} &= \lim_{\delta \to 0} \frac{\int_{\boldsymbol{t} \in \mathbb{R}^n} \mathbf{1}_{h(\boldsymbol{t}+\boldsymbol{x})=k} \left( v(\boldsymbol{t} - \delta \hat{\phi}) - v(\boldsymbol{t}) \right) d\boldsymbol{t}}{\delta} \\
&= \lim_{\delta \to 0} \int_{\boldsymbol{t} \in \mathbb{R}^n} \frac{\mathbf{1}_{h(\boldsymbol{t}+\boldsymbol{x})=k} \left( v(\boldsymbol{t} - \delta \hat{\phi}) - v(\boldsymbol{t}) \right)}{\delta} d\boldsymbol{t} \\
&= \lim_{\delta \to 0} \int_{\boldsymbol{t} \in \mathbb{R}^n} \mathbf{1}_{h(\boldsymbol{t}+\boldsymbol{x})=k} \frac{v(\boldsymbol{t} - \delta \hat{\phi}) - v(\boldsymbol{t})}{\delta} d\boldsymbol{t} \\
&= \int_{\boldsymbol{t} \in \mathbb{R}^n} \mathbf{1}_{h(\boldsymbol{t}+\boldsymbol{x})=k} \lim_{\delta \to 0} \frac{v(\boldsymbol{t} - \delta \hat{\phi}) - v(\boldsymbol{t})}{\delta} d\boldsymbol{t} \\
&= - \int_{\boldsymbol{t} \in \mathbb{R}^n} \mathbf{1}_{h(\boldsymbol{t}+\boldsymbol{x})=k} \lim_{\delta \to 0} \frac{v(\boldsymbol{t}) - v(\boldsymbol{t} - \delta \hat{\phi})}{\delta} d\boldsymbol{t} \\
&= - \int_{\boldsymbol{t} \in \mathbb{R}^n} \mathbf{1}_{h(\boldsymbol{t})=k} \lim_{\delta \to 0} \frac{v(\boldsymbol{t} - \boldsymbol{x}) - v(\boldsymbol{t} - \boldsymbol{x} - \delta \hat{\phi})}{\delta} d\boldsymbol{t} \\
&= \int_{\boldsymbol{t} \in \mathbb{R}^n} \mathbf{1}_{h(-\boldsymbol{t}+\boldsymbol{x})=k} \lim_{\delta \to 0} \frac{v(-\boldsymbol{t}) - v(-\boldsymbol{t} - \delta \hat{\phi})}{\delta} d\boldsymbol{t} \\
&= - \int_{\boldsymbol{t} \in \mathbb{R}^n} \mathbf{1}_{h(-\boldsymbol{t}+\boldsymbol{x})=k} \lim_{\delta \to 0} \frac{v(\boldsymbol{t} + \delta \hat{\phi}) - v(\boldsymbol{t})}{\delta} d\boldsymbol{t} \\
&= - \int_{\boldsymbol{t} \in \mathbb{R}^n} \mathbf{1}_{h(\boldsymbol{t}+\boldsymbol{x})=k} \left( \nabla v(\boldsymbol{t}) \cdot \hat{\phi} \right) d\boldsymbol{t}
\end{aligned}
\tag{48}
$$

The directional derivative of $\mu_k$ is maximised when the binary function takes 1 if and only if $\nabla v(\boldsymbol{t}) \cdot \phi < 0$. Also, $v$ is even in every dimension, and when $\nabla v(\boldsymbol{t}) \cdot \phi < 0$, it is necessary that if $\nabla v(-\boldsymbol{t}) \cdot \phi > 0$. Thus, the direction that maximises the directional derivative of $\mu_k$ can be expressed as

$$
\phi^* = - \begin{bmatrix} \int_{\boldsymbol{t} \in \mathbb{R}^n} \left| \frac{\partial v(\boldsymbol{t})}{\partial t_1} \right| d\boldsymbol{t} \\ \int_{\boldsymbol{t} \in \mathbb{R}^n} \left| \frac{\partial v(\boldsymbol{t})}{\partial t_2} \right| d\boldsymbol{t} \\ \vdots \\ \int_{\boldsymbol{t} \in \mathbb{R}^n} \left| \frac{\partial v(\boldsymbol{t})}{\partial t_n} \right| d\boldsymbol{t} \end{bmatrix} \tag{49}
$$

Thus, the upper bound of the directional derivative of $\mu_k$ can be written as

$$
\begin{aligned}
\nabla \mu_k(\boldsymbol{x}) \cdot \frac{\phi}{|\phi|} &\leq \frac{1}{2} \int_{\boldsymbol{t} \in \mathbb{R}^n} \left| \nabla v(\boldsymbol{t}) \cdot \frac{\phi^*}{|\phi^*|} \right| d\boldsymbol{t} \\
&= \frac{1}{2 |\phi^*|} \int_{\boldsymbol{t} \in \mathbb{R}^n} \left| \sum_{i=1}^n \left( \frac{\partial v(\boldsymbol{t})}{\partial t_i} \int_{\boldsymbol{\tau} \in \mathbb{R}^n} \left| \frac{\partial v(\boldsymbol{\tau})}{\partial \tau_i} \right| d\boldsymbol{\tau} \right) \right| d\boldsymbol{t}
\end{aligned}
\tag{50}
$$

Since function $v$ is not a delta distribution PDF, this bound is always a finite number. $\qquad\square$

## B.7 Example of Corollary 3.5

*Example* 1.

$$
\begin{aligned}
&= \frac{1}{2\,|\phi^*|} \int_{\boldsymbol{t}\in\mathbb{R}^n} \left| \sum_{i=1}^n \left( \frac{\partial v(\boldsymbol{t})}{\partial t_i} \int_{\boldsymbol{\tau}\in\mathbb{R}^n} \left| \frac{\partial v(\boldsymbol{\tau})}{\partial \tau_i} \right| d\boldsymbol{\tau} \right) \right| d\boldsymbol{t}. \\
&= \frac{1}{2} \int_{\boldsymbol{t}\in\mathbb{R}^n} \left| \sum_{i=1}^n \left( \frac{\partial v(\boldsymbol{t})}{\partial t_i} \right) \right| d\boldsymbol{t}. \\
&= \frac{1}{2} \int_{\boldsymbol{t}\in\mathbb{R}^n} \left| \left( \frac{\partial v(\boldsymbol{t})}{\partial t_0} \right) \right| d\boldsymbol{t}. \\
&= \frac{1}{2} \int_{\boldsymbol{t}\in\mathbb{R}^n} \left| \left( \frac{\partial v(\boldsymbol{t})}{\partial t_0} \right) \right| d\boldsymbol{t}.
\end{aligned}
\tag{51}
$$

Suppose we have input $\boldsymbol{x} \in \mathbb{R}$, $v : \mathbb{R} \to \mathbb{R}$, and $v$ is a symmetric uniform distribution function. According to Corollary 3.5, the slope of $\mu_k$ in this example is within $\pm\frac{1}{2\epsilon}$. We can validate this value in Equation (52) using Leibniz's rule for differentiation under the integral sign.

$$
\frac{d(\mathbf{1}_{h()=k} * v)(\boldsymbol{x})}{d\boldsymbol{x}} = \frac{1}{2\epsilon} \frac{d}{d\boldsymbol{x}} \int_{-\epsilon}^{\epsilon} \mathbf{1}_{h(\boldsymbol{x}-\boldsymbol{t})=k}\, d\boldsymbol{t} = \frac{1}{2\epsilon} \left( \mathbf{1}_{h(\boldsymbol{x}-\epsilon)=k} - \mathbf{1}_{h(\boldsymbol{x}+\epsilon)=k} \right)
\tag{52}
$$

The intuition of this example is that when a vicinity shifts from a region where all samples are labelled with one class to a region where all samples are labelled with another class, the slope reaches its maximum.

## B.8 Proof of Corollary 3.6

**Corollary 3.6.** *If $h^*$ is optimal for the probabilistic robustness with respect to an $L^\infty$-vicinity on a given distribution, then the vicinity size for the deterministically robust region around each consistent is $\epsilon \left( 1 - (2\kappa)^{\frac{1}{n}} \right)$.*

*Proof.* An $L^\infty$ norm looks like a $n$-dimensional cube. A two-dimensional illustration is given in Figure 1d. Generally, the vicinity function can be expressed as

$$
v(\boldsymbol{x}) = \begin{cases} (2\epsilon)^{-n} & \text{for } \|\boldsymbol{x}\|_\infty \le \epsilon \\ 0 & \text{otherwise.} \end{cases}
\tag{53}
$$

According to Theorem 3.4, if we would like to find the closest adversarial example to a consistent input at the upper right centre of the yellow vicinity, we first need to find a shift magnitude that causes as large as a $\mu_{k^*}$ drop by $1/2 - \kappa$. Suppose this drop goes further and the closest (probabilistically) consistent input is found. In this way, the minimum $\mu_k$ drop from consistent point $\boldsymbol{x}$ and its (probabilistically) consistent adversarial example is $1 - \kappa - \kappa = 1 - 2\kappa$ In an $L^\infty$ vicinity scenario, the magnitude of shift can be solved based on Equation (54).

$$
1 - 2\kappa = 1 - \min_{\hat{\phi}} \prod_{i=1}^n \max(0, 2\epsilon - \phi\hat{\phi}_i)
\tag{54}
$$

where $\hat{\phi}_i$ is each element of the unit directional vector $\hat{\phi}$. Since the shift is within the vicinity, we can write $\max(0, 2\epsilon - \phi\hat{\phi}_i)$ as $2\epsilon - \phi\hat{\phi}_i$. For two identical n-dimensional cubes, the fastest way to reduce the overlap is to move one of them in the diagonal direction, such that $\hat{\phi}_i = \frac{1}{\sqrt{n}}$. Then the overlap volume becomes $(2\epsilon - \phi/\sqrt{n})^n$. Then, Equation (54) can be simplified as $2\kappa(2\epsilon)^n = (2\epsilon - |\phi|/\sqrt{n})^n$. Solving this equation, we get $\epsilon(1 - (2\kappa)^{\frac{1}{n}})$. This $\epsilon(1 - (2\kappa)^{\frac{1}{n}})$ also serves as the vicinity size for deterministic robustness at this input. As $n$ grows, this vicinity size decreases. As $\kappa$ grows, this vicinity size may grow. $\qquad\square$

We may visualise the effect of Corollary 3.6 using a two-dimensional example in Figure 1d. Let $\boldsymbol{x}, \boldsymbol{x}'$ be two probabilistically consistent inputs and $h(\boldsymbol{x}) \ne h(\boldsymbol{x}')$. Suppose $\boldsymbol{x}'$ is the nearest adversarial

example (of $\boldsymbol{x}$) that achieves its own probabilistic consistency. Thus, $\boldsymbol{x}'$ must be in the direction (that travels away from $\boldsymbol{x}$) that fastest decreases the vicinity overlap (suggested by Theorem 3.4), *i.e.*, the diagonal (suggested by Corollary 3.6). Consequently, the nearest adversarial example of $\boldsymbol{x}$ would occur on the midpoint between $\boldsymbol{x}$ and $\boldsymbol{x}'$. The shift from $\boldsymbol{x}$ to $\boldsymbol{x}'$ is $-2(\phi_1\hat{\boldsymbol{x_1}} + \phi_2\hat{\boldsymbol{x_2}})$. Each triangle accounts for a $\kappa$ portion of the original vicinity volume, and the vicinity overlap is $2\kappa$. The dashed box $\mathbb{V}^{\downarrow\kappa}$ has side length $2\phi_i$. Thus, solving $(2\epsilon - 2\phi_i)^2 = 2\kappa(2\epsilon)^2$, we get $\mathbb{V}^{\downarrow\kappa}$ has vicinity size $\phi_1 = \phi_2 = (1 - \sqrt{2\kappa})\epsilon$. Although Corollary 3.6 specifically captures the $L^\infty$ scenario, we remark that other vicinity types can be analysed similarly. First, the direction that fastest decreases the vicinity overlap needs to be found. Then, the distance between the input and its nearest adversarial example can be measured.

### B.9 Extended Proof of Corollary 3.8

**Theorem 3.8.** *The upper bound of probabilistic robust accuracy monotonically increases as $\kappa$ grows. Further, for all tolerance levels $\kappa$, the upper bound of probabilistic robust accuracy lies between the upper bound of deterministic robust accuracy and vanilla accuracy. Formally,*

$$\forall \kappa_1, \kappa_2. \quad (\kappa_1 < \kappa_2) \to \min_h \Upsilon_{\text{prob}}^+(D, h, \mathcal{V}, \kappa_1) \le \min_h \Upsilon_{\text{prob}}^+(D, h, \mathcal{V}, \kappa_2)$$
$$\forall \kappa. \quad \min_h \Upsilon_{\text{rob}}^+(D, h, \mathbb{V}) \le \min_h \Upsilon_{\text{prob}}^+(D, h, \mathcal{V}, \kappa) \le \min_h \Upsilon_{\text{acc}}^+(D, h) \tag{55}$$

*Proof.* Let $\kappa_1 < \kappa_2$, such that for $\kappa_1, \kappa_2$, we have their corresponding error as the following equation.

$$e(\boldsymbol{x}, \kappa_1) - e(\boldsymbol{x}, \kappa_2) = 1 - u\left(\kappa_1 - 1 + \sum_{k=0}^{K-1} \mu_k(\boldsymbol{x})\,\mathbf{1}_{h(\boldsymbol{x})=k}\right)\left(\sum_{y=0}^{K-1} P(\mathrm{y} = y | \mathrm{x} = \boldsymbol{x})\,\mathbf{1}_{h(\boldsymbol{x})=y}\right)$$
$$- 1 + u\left(\kappa_2 - 1 + \sum_{k=0}^{K-1} \mu_k(\boldsymbol{x})\,\mathbf{1}_{h(\boldsymbol{x})=k}\right)\left(\sum_{y=0}^{K-1} P(\mathrm{y} = y | \mathrm{x} = \boldsymbol{x})\,\mathbf{1}_{h(\boldsymbol{x})=y}\right) \tag{56}$$

Note the cancelled 1 and the like terms in the rest of the terms, we can further write the above equation as Equation (57), where we let $T = \sum_{k=0}^{K-1} \mu_k(\boldsymbol{x})\,\mathbf{1}_{h(\boldsymbol{x})=k}$. Note that $T$ is just a temporary substituting variable, and we do not mean to use it to denote any particular quantity.

$$e(\boldsymbol{x}, \kappa_1) - e(\boldsymbol{x}, \kappa_2) = \left(\sum_{y=0}^{K-1} P(\mathrm{y} = y | \mathrm{x} = \boldsymbol{x})\,\mathbf{1}_{h(\boldsymbol{x})=y}\right)\left(u\left(\kappa_2 - 1 + T\right) - u\left(\kappa_1 - 1 + T\right)\right) \tag{57}$$

Now that $e(\boldsymbol{x}, \kappa_1) - e(\boldsymbol{x}, \kappa_2)$ is a product of two expressions. Since the posterior probability $P(\mathrm{y} \mid \mathrm{x} = \boldsymbol{x})$ is non-negative, the sum of posteriors, *i.e.*, the former expression, is non-negative. Thus, the sign of $e(\boldsymbol{x}, \kappa_1) - e(\boldsymbol{x}, \kappa_2)$ is the same as $u\left(\kappa_2 - 1 + T\right) - u\left(\kappa_1 - 1 + T\right)$.

Since $\kappa_1 < \kappa_2$, we get $\kappa_1 - 1 + T < \kappa_2 - 1 + T$. Further, the unit step function ($u$) is monotonically increasing, we get $e(\boldsymbol{x}, \kappa_1) - e(\boldsymbol{x}, \kappa_2) \ge 0$. A more stringent $\kappa$ (*i.e.*, smaller) leads to a lower or equal upper bound of probabilistic robust accuracy. For deterministic robust accuracy, $\kappa = 0$, which is the least value. Thus, deterministic robust accuracy has a lower upper bound than probabilistic robust accuracy does.

On the other hand, from the intuition of error combination in Section 3.1, we get that the combined error is $e(\boldsymbol{x}) = 1 - (1 - e_{\text{cns}}(\boldsymbol{x}, h; \mathcal{V}, \kappa))(1 - e_{\text{cor}}(\boldsymbol{x}, h; P(\mathrm{y} \mid \boldsymbol{x})))$. As $0 \le e_{\text{cns}} \le 1$, we get $e(\boldsymbol{x}) \ge 1 - (1 - e_{\text{cor}}(\boldsymbol{x}, h; \kappa))$. Note that the expectation of $e_{\text{cor}}(\boldsymbol{x}, h; \kappa)$ is the error in accuracy. Thus, $\Upsilon_{\text{prob}}^+(D, h, \mathcal{V}, \kappa) \le \Upsilon_{\text{acc}}^+(D, h)$. So it is with their upper bounds. $\square$

## C  Additional Experiments and Plots

In this section, we present the results of some additional experiments in which we investigate the effect of the upper bounds and decision rules.

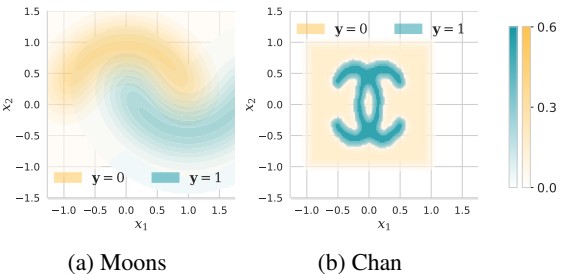

(a) Moons        (b) Chan

Figure 5: This figure illustrates the conditional distribution for (a) Moons (Pedregosa et al., 2011) and (b) Chan (Chen et al., 2023).

Table 2: Probabilistic robustness of classifiers before voting. The value in parentheses represents the 95% confidence level range when we repeat the same program 100 times.

|  | Moons | Chan | FashionMNIST | CIFAR-10 |
|---|---|---|---|---|
| DA (Shorten & Khoshgoftaar, 2019) | 85.35 ($\pm$0.020%) | 67.96 ($\pm$0.023%) | 84.12 ($\pm$0.000%) | 76.07 ($\pm$0.021%) |
| RS (Cohen et al., 2019) | 84.76 ($\pm$0.020%) | 64.67 ($\pm$0.002%) | 86.29 ($\pm$0.012%) | 87.98 ($\pm$0.038%) |
| CVaR (Robey et al., 2022) | 85.52 ($\pm$0.017%) | 69.46 ($\pm$0.034%) | 88.50 ($\pm$0.028%) | 90.63 ($\pm$0.007%) |

## C.1 SETUP DETAILS

The experiments are conducted with four data sets: two synthetic ones (*i.e.*, Moons and Chan (Chen et al., 2023), whose distributions are illustrated in Figure 5) and two standard benchmarks (*i.e.*, FashionMNIST (Xiao et al., 2017) and CIFAR-10 (Krizhevsky et al., 2009)). Moons is used for binary classification with two-dimensional features, where each class's distribution is described analytically with specific likelihood equations, and uses a three-layer Multi-Layer Perceptron (MLP) neural network for classification. The Chan data set, also for binary classification with two-dimensional features, differs in that it does not follow a standard PDF pattern, requiring kernel density estimation (KDE) for non-parametric PDF estimation, and also uses the three-layer MLP. FashionMNIST, a collection of fashion item images, involves a 10-class classification task with 784-dimensional inputs (28×28 pixel grayscale images). Each class has an equal prior probability, and their conditional distributions are estimated non-parametrically using KDE. CIFAR-10 uses images with a resolution of 32×32 pixels. Similar to FashionMNIST, it has a balanced class distribution and is estimated using KDE. We use a seven-layer convolutional neural network (CNN-7) (Shi et al., 2021) as the classifier of both FashionMNIST and CIFAR-10. We adopt a direct approach (Ishida et al., 2023; Zhang & Sun, 2024) to compute the original Bayes error and deterministic robustness Bayes error of both real-world data sets (Ishida et al., 2023). We consider each class posterior may contain some noise $\xi_i$, such that the labels are $\left\{ p(y_i \mid \boldsymbol{x}_i) + \xi_i \right\}_i^n$, and $0 \le p(y_i|\boldsymbol{x}_i) + \xi_i \le 1$, $\mathrm{E}[\xi_i|y_i] = 0$. Then, with the category of each example $y_i$, we can get the unbiased estimator of Bayes error (Ishida et al., 2023). For the training algorithms of $h$ from randomised smoothing (RS, Cohen et al., 2019), we follow Cohen et al. (2019) and Lecuyer et al. (2019), and add Gaussian noise to the training samples, with the noise level is set at 0.5 of the total range. When inference is needed, the number of samples is 1,000 and the confidence parameter is $\alpha = 0.01$.

## C.2 STATISTICAL SIGNIFICANCE OF EXPERIMENTS

We include confidence levels of performances of trained classifiers Table 2 and their corresponding voting classifiers Table 3. We provide their statistical significance to support the claims addressed by our research questions.

Besides, the train/validation splits follow that of the data set (Xiao et al., 2017; Krizhevsky et al., 2009) if there is already a split guideline. For Moons (Pedregosa et al., 2011) and Chan (Chen et al., 2023), we follow the setup in (Zhang & Sun, 2024).

Table 3: Probabilistic robustness of classifiers after voting. The value in parentheses represents the 95% confidence level range when we repeat the same program 100 times.

| | Moons | Chan | FashionMNIST | CIFAR-10 |
|---|---|---|---|---|
| DA (Shorten & Khoshgoftaar, 2019) | 85.60 (±0.034%) | 68.86 (±0.006%) | 87.48 (±0.022%) | 81.38 (±0.026%) |
| RS (Cohen et al., 2019) | 85.18 (±0.005%) | 66.77 (±0.008%) | 88.13 (±0.037%) | 88.95 (±0.008%) |
| CVaR (Robey et al., 2022) | 85.66 (±0.011%) | 70.05 (±0.005%) | 91.07 (±0.028%) | 90.77 (±0.011%) |

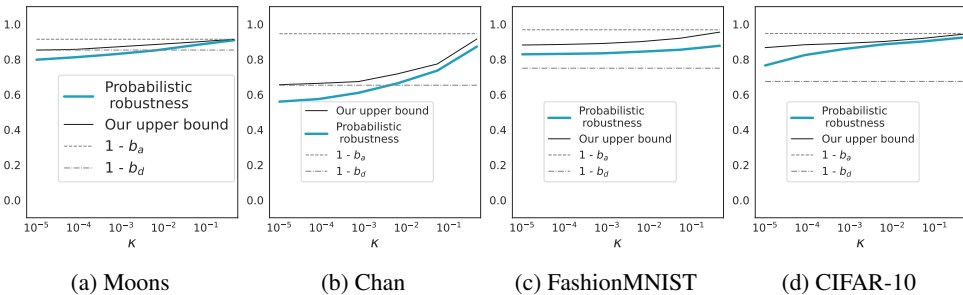

| (a) Moons | (b) Chan | (c) FashionMNIST | (d) CIFAR-10 |
|---|---|---|---|

Figure 6: An alternative illustration of Figure 3 with a logarithmic scale on $\kappa$. As $\kappa$ increases from $10^{-5}$ to $10^{-1}$ (in the small value region), we plot the upper bounds of probabilistic robust accuracy as well as classifiers' probabilistic robust accuracy change. The upper bounds of vanilla accuracy (Ishida et al., 2023) and the deterministic robust accuracy (Zhang & Sun, 2024) are also included as references.

### C.3 How does the sample size affect the voting effectiveness?

The voting process of classifier $h^\dagger$ can be viewed as a process of taking the expected value of prediction within the vicinity. The law of large numbers is key to understanding the relationship between sample size and expectation. It states that as the sample size increases, the sample mean converges to the true expectation.

To validate this effect empirically, as well as to verify a suitable sample size at which the performance of the voting classifiers can be properly represented, we gradually increase the sample size from 10 to 10,000. As demonstrated in Figure 7, at a small sample size, the voting may result in an increase or a decrease in the performance. However, as the sample exceeds 100, its positive impact on the probabilistic robust accuracy becomes more noticeable. Thus, our empirical intuition matches Theorem 3.2.

## D   More Detailed Explanation of Bayes Error on Deterministic Robustness and Vanilla Accuracy

### D.1   Bayes Error on Deterministic Robustness

Prior work shows that when all examples in the original distribution concurrently assign labels to their respective vicinity, the effect can be seen as convolving this given distribution with the vicinity. This convolved distribution represents optimisation towards robustness-aware training, as captured by Theorem D.1 (Theorem 1 in (Zhang & Sun, 2024)).

**Theorem D.1.** *Given a distribution $D$ for classification, deterministic robustness-aware training does not optimise the classifiers to fit $D$. Rather, it optimises classifiers towards $D * v$, i.e., convolved distribution between $D$ and vicinity $v(\boldsymbol{x})$.*

Hereafter, the optimal robust classifier can be approximated by the Bayes classifier of $D' = D * v$. For a given vicinity $\mathbb{V}$, the irreducible robustness error of function $h$ on distribution $D$ would thus be given as follows, where $p$ denotes the PDF in $D'$.

$$\min \Upsilon_{\text{rob}}^-(D, h, ) = \int_{\text{near boundary}} p(\boldsymbol{x}) d\boldsymbol{x} + \int_{\neg \text{near boundary}} \left(1 - \max_k p(\mathbf{y} = k | \mathbf{x} = \boldsymbol{x})\right) p(\boldsymbol{x}) d\boldsymbol{x} \quad (58)$$

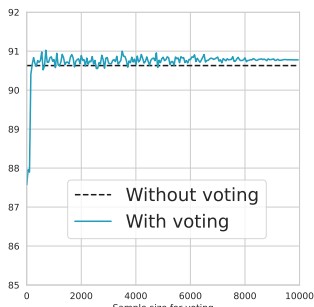

Figure 7: Probabilistic robust accuracy of voting classifier (CVaR (Robey et al., 2022) tested on CIRAR-10 (Krizhevsky et al., 2009)) as the sample size grows to 10,000.

Intuitively, when an input point $\boldsymbol{x}$ is not near the boundary, the irreducible robustness error contributed from this point would follow the standard Bayes error computation, *i.e.*, $(p(\boldsymbol{x}) - \max_k p(\mathrm{y} = k, \mathbf{x} = \boldsymbol{x}))$, as captured in the second term in Equation (58). When an input is near the boundary, at least one different prediction is in its vicinity. In this case, this point contributes $p(\boldsymbol{x})$ to the irreducible robustness error, which is captured in the first term in Equation (58).

To determine the subset of $\mathbb{X}$ where an input point is near the boundary, we can "harden" distribution $D'$ and convolve it with $v$ (Zhang & Sun, 2024). Formally, let $\lceil D' \rceil$ denote a distribution with the following posterior distribution, where a draw of maximum posterior may be broken arbitrarily.

$$p(\mathrm{y} = k | \boldsymbol{x}) = \begin{cases} 1 & \text{if in } D', \ p_(\mathrm{y} = k | \boldsymbol{x}) = \max_{k'} p(\mathrm{y} = k' | \boldsymbol{x}) \\ 0 & \text{if in } D', \ p_(\mathrm{y} = k | \boldsymbol{x}) \neq \max_{k'} p(\mathrm{y} = k' | \boldsymbol{x}) \end{cases} \tag{59}$$

Next, we can represent the domain near the boundary using the posterior distribution of $D^{\dagger}$ where $D^{\dagger} = \lceil D' \rceil * v$, as shown in the following equations.

$$\mathbb{K}_{D^{\dagger}} = \{ \boldsymbol{x} \mid (\mathbf{x}, \mathrm{y}) \sim D^{\dagger}, \max_k p(k | \mathbf{x} = \boldsymbol{x}) < 1 \}$$
$$\mathbb{X} \setminus \mathbb{K}_{D^{\dagger}} = \{ \boldsymbol{x} \mid (\mathbf{x}, \mathrm{y}) \sim D^{\dagger}, \max_k p(k | \mathbf{x} = \boldsymbol{x}) = 1 \} \tag{60}$$

Therefore, the Bayes error for deterministic robustness of $D$ is the Bayes error of $D * v$ plus the joint probability of non-max classes in $\mathbb{K}_{D^{\dagger}}$. We may rewrite Equation (58) in the following.

$$\begin{aligned} \min \Upsilon_{\mathrm{rob}}^{-}(D, h,) &= \int_{\mathbb{K}_{D^{\dagger}}} p(\boldsymbol{x}) d\boldsymbol{x} + \int_{\mathbb{X} \setminus \mathbb{K}_{D^{\dagger}}} \left( 1 - \max_k p(\mathrm{y} = k | \mathbf{x} = \boldsymbol{x}) \right) p(\boldsymbol{x}) d\boldsymbol{x} \\ &= \int_{\mathbb{K}_{D^{\dagger}}} 1 \cdot p(\boldsymbol{x}) d\boldsymbol{x} + \int_{\mathbb{X} \setminus \mathbb{K}_{D^{\dagger}}} \left( 1 - \max_k p(k | \boldsymbol{x}) \right) p(\boldsymbol{x}) d\boldsymbol{x} \\ &= \int_{\mathbb{K}_{D^{\dagger}}} 1 \cdot p(\boldsymbol{x}) d\boldsymbol{x} + \int_{\mathbb{X} \setminus \mathbb{K}_{D^{\dagger}}} 1 \cdot p(\boldsymbol{x}) - \left( \max_k p(k | \boldsymbol{x}) \right) p(\boldsymbol{x}) d\boldsymbol{x} \\ &= \int_{\mathbb{X}} 1 \cdot p(\boldsymbol{x}) d\boldsymbol{x} - \int_{\mathbb{X} \setminus \mathbb{K}_{D^{\dagger}}} \left( \max_k p(k | \boldsymbol{x}) \right) p(\boldsymbol{x}) d\boldsymbol{x} \\ &= \int_{\mathbb{X}} 1 \cdot p(\boldsymbol{x}) d\boldsymbol{x} - \int_{\mathbb{X}} \mathbf{1}_{\boldsymbol{x} \notin \mathbb{K}_{D^{\dagger}}} \left( \max_k p(k | \boldsymbol{x}) \right) p(\boldsymbol{x}) d\boldsymbol{x} \\ &= \int_{\mathbb{X}} 1 \cdot p(\boldsymbol{x}) - \mathbf{1}_{\boldsymbol{x} \notin \mathbb{K}_{D^{\dagger}}} \left( \max_k p(k | \boldsymbol{x}) \right) p(\boldsymbol{x}) d\boldsymbol{x} \\ &= \int_{\mathbb{X}} \left( 1 - \mathbf{1}_{\boldsymbol{x} \notin \mathbb{K}_{D^{\dagger}}} \max_k p(k | \boldsymbol{x}) \right) p(\boldsymbol{x}) d\boldsymbol{x} \end{aligned} \tag{61}$$

Since this is the integration regarding the distribution $D'$, we can rewrite the integration form back into the expected value form, *i.e.*, Equation (7).

