# OpenReview forum: "Are Probabilistic Robust Accuracy Bounded"
_ICLR.cc/2025/Conference — Submitted to ICLR 2025_

### Official Review · Reviewer_evQe · 2024-11-03

**Soundness:** 3
**Presentation:** 3
**Contribution:** 3
**Rating:** 6
**Confidence:** 2

**Summary:**

The paper studies probabilistic robust classifiers (PRC) as defined in Robey et al. (2022). It derives an expression of the Bayes error that bounds the PRC. They further, show that this bound is bounded between the Bayes error of the vanilla classifier and the deterministically robust Bayes error. They evaluate the PRC accuracies and Bayes errors empirically.

**Strengths:**

S1: The paper is well structured and actively helps the user to get through it. It often announces next steps and summarizes theorems and key insights in plain language. That helps tremendously in parsing the paper. E.g. L203-L207

S2: The paper is a natural extension of previous work to probabilistic robust classifiers. Understanding their properties is an important endeavor. The result does not come as a surprise ("accuracy vs robustness guarantee" trade-off is well established), but it is original work and a worthwhile contribution to the robust ML community, furthering our understanding of these models.

S3: The authors provide the code for their experiments.

**Weaknesses:**

W1: The experimental setup is on the light side.

- SVHN, CIFAR-100, TinyImageNet (and similar datasets) would be appreciated. This experimental setup mirrors existing work, so I dont think this is a reason for rejection but, it would be useful to see this validated across a more diverse set of datasets.
- The range of $\kappa$ values shown leaves out valuable insights into values $<< 0.1$. I would suggest, e.g. to replicate Figure 3 with logscale x axis, e.g. starting with 1e-5. How do the Bayes errors compare for such small $\kappa$? How does the prob. robust classifier bound approach the bound of the deterministic classifier when $\kappa \rightarrow 0$?
I would suggest the authors include more $\kappa$ values in this work and more datasets / architectures in a revision or future works section.

W2: L176-190 (in particular L186-L188) are hard to follow when not familiar with Zhang & Sun's work. A more detailed explanation would be appreciated. For instance, the authors could keep it short in the main body but add a short Appendix with more details.

W3: The work would benefit from a thorough proof-reading. For example (not complete list):

- The title is grammatically incorrect. Should probably be: "Probabilistic robust accuracy is bounded".
- Comma after "classses" incorrect (L178), "statistiCAL significance" (L425)
- The language throughout the paper could be more nuanced. For instance, "The fundamental problem in this study" (L194) is not clear. It is the "problem studied in this work", not the "problem of the work". "unbearable" in the abstract is too colloquial.
- L224 should be $\mu_k(x) \triangleq$ ...
- "whereas" in line 536 is the wrong word.
I suggest the authors proofread the paper carefully.

**Minor**

MW1: The subscripts for the errors in Section 3.1 are only explained long after introducing them (L251 to be precise). Moving up that section and explaining subscripts before presenting them can save the reader some time wondering what $e_{cor}$ in Eq 9 and $e_{cns}$ ins Eq 11 are.

MW2: The authors don't "compute" the voting classifier, they estimate it L(428-L429).

**Questions:**

Q1: The voting classifier comes out of nowhere in L428. What is the rationale behind presenting it there?

Q2: What are the parameters for cohen RS? Number of samples etc, threshold for abstaining etc. (L453)

Q3: The authors seem surprised that the accuracy is bounded below the Bayes error, i.e. the authors state "It remains an open question whether an upper limit on accuracy exists when optimizing for probabilistic robustness" (L015-L017). To me it seems trivial that a Bayes classifier exists and that it bounds the accuracy of any other the classifier (very much by the definition of the Bayes classifier). I value the contribution of deriving _what_ it is, but I struggle to understand why it was debatable _whether_ it exists. Could the authors please clarify this point?

Q4: "Deterministic robustness requires a zero probability of adversarial samples..." (L135). This is necessary but not sufficient for determinsitic robustness, in my opinion. If there exists an adversary in the neigbourhood of $x$ with measure 0, the classifier is not deterministically robust.

Q5: Why are 50% of adversarial samples in the vicinity considered acceptable for Cohen et al. (2019)? (see L144). The authors list Gaussian Smoothing under "probabilistic" robustness. In my opinion, this is incorrect. The classifier itself is _deterministically robust_ and the probabilistic nature comes in as we need to approximate the true classifier, not from the classiifer itself. Is there any literature that proves that e.g. Eq 5 holds for the empirical approximation to the smoothed classifier for some reasonable $\mathcal{V}(x)$?

Q: Figure 2 and 3 contains estimates of the Bayes errors to bound the accuracies above. How precise is this estimate? Is it computationally tractable to compute this on CIFAR-100, TinyImageNet or similar datasets?

---

> ### Author Response · Authors · 2024-11-21
> **Response to Reviewer evQe (Part 1/2)**
>
> We thank the reviewer for the time and effort in working on the review and for the kind words on our work. We carefully address the concerns as follows. If any concerns still persist, we are more than happy to discuss them and edit our paper accordingly.
>
> **Regarding the comment: “W1: The experimental setup is on the light side.
> SVHN, CIFAR-100, TinyImageNet (and similar datasets) would be appreciated. This experimental setup mirrors existing work, so I don't think this is a reason for rejection but, it would be useful to see this validated across a more diverse set of datasets.**
>
> **The range of $\kappa$ values shown leaves out valuable insights into values $\ll 1$. I would suggest, e.g. to replicate Figure 3 with logscale x axis, e.g. starting with 1e-5. How do the Bayes errors compare for such small $\kappa$? How does the prob. robust classifier bound approach the bound of the deterministic classifier when $\kappa\to 0$? I would suggest the authors include more $\kappa$ values in this work and more datasets/architectures in a revision or future works section.”**
>
> Great suggestion. We have now managed to extend the experimental evaluation to the suggested datasets. The upper bound of probabilistic robust accuracy and the state-of-the-art performance of SVHN and TinyImageNet are shown in the following table. CIFAR-100 is still running, and we are happy to add the results in our final version.
>
>
> |             | SVHN ($\kappa = 0.1$, $\epsilon = 2/255$) | TinyImageNet ($\kappa = 0.1$, $\epsilon = 2/255$) |
> |-------------|-------------------------------------|---------------------------------------------|
> | Upper bound |                97.02                |                    84.11                    |
> | CVaR        |                93.03                |                    54.38                    |
>
> Thank you for pointing this out! As you suggested, the change of the upper bound of probabilistic robust accuracy against the growth of $\kappa$ is shown in log-scaled $\kappa$ in Figure 6 in the revised version (Section C). Also, we are happy to include it in the main body if needed.
>
> **Regarding the comment: “W2: L176-190 (in particular L186-L188) are hard to follow when not familiar with Zhang & Sun's work. A more detailed explanation would be appreciated. For instance, the authors could keep it short in the main body but add a short Appendix with more details.”**
>
> Thanks for the suggestion. We have added Appendix D in the revised version to help with understanding.
>
> **Regarding the comment: W3 and minor**
>
> Thanks for pointing them out. We have fixed them in the revised version and added abbreviations in **Line 216** and **Line 232** to denote the subscripts of Eq (9) and Eq (11).

---

> > ### Author Response · Authors · 2024-11-21
> > **Response to Reviewer evQe (Part 2/2)**
> >
> > **Regarding the comment: “Q1: The voting classifier comes out of nowhere in L428. What is the rationale behind presenting it there?”**
> >
> > Thanks for the question. This voting classifier comes from the rationale of Theorem 3.2 on page 5. We have made the connection clearer in the revised version (in **Line 428**). Specifically, Theorem 3.2 suggests in the optimal condition for probabilistic robustness, each prediction matches the voted prediction in the vicinity. Following this rationale, this research question tries to verify if voting indeed improves classification robustness, even if a classifier is only close to optimal.
> >
> > **Regarding the comment: “Q2: What are the parameters for cohen RS? Number of samples etc, threshold for abstaining etc. (L453)”**
> >
> > Thanks for the question. We revised the draft and provided the configurations for RS at **Line 1226**. In short, for each distribution, the noise level is set at 0.5 of the total range. We only adopt RS training which does not involve a threshold for abstaining etc.
> >
> >
> > **Regarding the comment: “Q3: The authors seem surprised that the accuracy is bounded below the Bayes error, i.e. the authors state "It remains an open question whether an upper limit on accuracy exists when optimizing for probabilistic robustness" (L015-L017). To me it seems trivial that a Bayes classifier exists and that it bounds the accuracy of any other the classifier (very much by the definition of the Bayes classifier). I value the contribution of deriving what it is, but I struggle to understand why it was debatable whether it exists. Could the authors please clarify this point?”**
> >
> > Thanks for the instructive comments. We have revised the writing accordingly to reduce the “surprisingness”, as follows: “It remains an open question what the upper limit on accuracy is when optimizing for probabilistic robustness”.
> >
> > **Regarding the comment: “Q4: "Deterministic robustness requires a zero probability of adversarial samples..." (L135). This is necessary but not sufficient for deterministic robustness, in my opinion. If there exists an adversary in the neighbourhood of  with measure 0, the classifier is not deterministically robust.”**
> >
> > Thank you for the constructive comments. That is right, and that is exactly what we mean, *i.e.*, this is a necessary condition for deterministic robustness. Having zero probability in a vicinity could already be a very rigorous and hard condition to achieve. Thus, we put this statement as a support that achieving deterministic robustness is challenging.
> >
> > **Regarding the comment: “Q5: Why are 50% of adversarial samples in the vicinity considered acceptable for Cohen et al. (2019)? (see L144). The authors list Gaussian Smoothing under "probabilistic" robustness. In my opinion, this is incorrect. The classifier itself is deterministically robust and the probabilistic nature comes in as we need to approximate the true classifier, not from the classifier itself. Is there any literature that proves that e.g. Eq 5 holds for the empirical approximation to the smoothed classifier for some reasonable?”**
> >
> > Thanks for the insightful comment. We agree that RS overall deterministically certifies a smoothed classifier. What we meant to say here is that the training method of RS improves the probabilistic robustness in the base classifier’s vicinity. Thus we take its training as a potential solution to probabilistic robustness.
> >
> > **50%**: Here we were trying to say that a small portion of adversarial examples is considered acceptable in probabilistic robustness. We do not mean that Cohen et al. [2019] consider 50% adversarial examples acceptable. We have revised the sentence and number in **Line 144** to avoid confusion.
> >
> >
> >
> >
> >
> >
> >
> >
> > **Regarding the comment: “Q: Figure 2 and 3 contains estimates of the Bayes errors to bound the accuracies above. How precise is this estimate? Is it computationally tractable to compute this on CIFAR-100, TinyImageNet or similar datasets?”**
> >
> > Thanks for the question. Comparing the analytically derived and numerically estimated result of the bounds, the error would be within 0.1%. For instance, on the Moons task, using the analytically derived closed-form distribution, the Bayes error for probabilistic robust accuracy is 10.13%. With a 600 $\times$ 600 grid for numerical estimation, the Bayes error is 10.14%.
> >
> > The proposed method is computationally tractable on TinyImageNet as well. The results that we have obtained now are shown in the table above.
> >
> > We truly value and appreciate your detailed comments on manuscript structuring, although a few of them currently might not have been fully addressed due to the page limit (*e.g.*, the subscript ‘cor’). Nevertheless, we are very willing to take your comments in our final version.

---

> > > ### Comment · Reviewer_evQe · 2024-11-27
> > >
> > > I thank the reviewer for their detailed rebuttal and the clarifications.
> > >
> > > A few comments:
> > >
> > > 1. I appreciate Figure 6 with smaller $\kappa$ values. The decay in performance for $\kappa \rightarrow 0$ is surprisingly low. I would like to suggest, to include the clean accuracy and the deterministic robust accuracy in Figure 3 and 6 to have reference.
> > >
> > > 2. The title asks whether probabilistic robust accuracy is bounded
> > >
> > > 4. I find the details on RS parameter insufficient (L1226) I would be interested in alpha, number of samples.
> > >
> > > I appreciate the authors' detailed rebuttal and revisions, which have clarified several aspects of the paper. While I am not ready to raise my score to an 8 at this time, there is an upwards tendancy. I will carefully consider the provided additions during the discussion with fellow reviewers and the Area Chair.

---

> > > > ### Author Response · Authors · 2024-11-27
> > > >
> > > > **Regarding the comment: “I appreciate Figure 6 with smaller $\kappa$ values. The decay in performance for $\kappa\to 0$ is surprisingly low. I would like to suggest, to include the clean accuracy and the deterministic robust accuracy in Figure 3 and 6 to have reference.”**
> > > >
> > > > Great suggestion. We have added upper bounds of clean accuracy and deterministic robust accuracy for reference in Figure 3 and Figure 6.
> > > >
> > > > **Regarding the comment: “The title asks whether probabilistic robust accuracy is bounded.”**
> > > >
> > > > Thank you for the comment. We have updated the title to “Probabilistic Robust Accuracy is Bounded” in the revised manuscript, though we might not be able to modify it on the OpenReview system at the moment.
> > > >
> > > >
> > > > **Regarding the comment: “I find the details on RS parameter insufficient (L1226) I would be interested in alpha, number of samples.”**
> > > >
> > > > Thanks for the constructive comments. The number of samples is 1,000 and the confidence parameter is $\alpha=0.01$ when an RS prediction is used. This is also updated in the revised manuscript in **Line 1230**.
> > > >
> > > >
> > > > Once again, we thank Reviewer for the time and effort in helping improve our paper. Should further clarification be required, feel free to ask, and we would be happy to respond.

---

### Official Review · Reviewer_1Vxa · 2024-11-04

**Soundness:** 3
**Presentation:** 4
**Contribution:** 3
**Rating:** 8
**Confidence:** 3

**Summary:**

This paper investigates the theoretical limits of probabilistic robust accuracy for neural networks in the face of adversarial attacks, using a Bayes error framework. The authors investigate whether probabilistic robustness can achieve a higher upper bound on robust accuracy than deterministic robustness, given the Bayes error, which represents the irreducible error inherent in classification tasks. The paper's findings are validated empirically across multiple datasets, demonstrating that probabilistic robustness is a better option in comparison to deterministic robustness.

**Strengths:**

- The paper presents a novel contribution by rigorously analyzing the upper bound of probabilistic robust accuracy from a Bayes error perspective.
- The work provides a rigorous theoretical analysis and the way the paper is written makes the mathematics easy to follow.
- The theoretical analysis and claims are properly supported by the experiments in the paper.

**Weaknesses:**

- I do not see any major weakness in the paper. Although, I do have a few questions regarding the experimental setup.

**Questions:**

- Could you clarify the choice of vicinity used in the experiments? If I understood correctly, the vicinity is defined using the $L^{\infty}$-norm. Are there other common choices for defining vicinity, and could evaluations be conducted with those as well?

- How do the upper and lower bounds in this work relate to the concept of certified radius, which is a significant measure of robustness in various probabilistic robustness methods, such as Randomized Smoothing?

---

> ### Author Response · Authors · 2024-11-20
> **Response to Reviewer 1Vxa**
>
> We thank the reviewer for the valuable comments and questions. Our response to each question is provided in turn below.
>
> **Regarding the comment: “Could you clarify the choice of vicinity used in the experiments? If I understood correctly, the vicinity is defined using the $L^\infty$ -norm. Are there other common choices for defining vicinity, and could evaluations be conducted with those as well?”**
>
> Thanks for your constructive comments. Indeed, we use the $L^\infty$-norm in our experiment. $L^2$-norm is also a common choice for defining vicinity and the evaluation can be conducted with $L^2$ and other vicinities as well. Similar to the $L^\infty$ case where the ratio of vicinity shrinking is $(1 − \sqrt{2\kappa})$ in two dimensions, the ratio for an $L^2$ in two dimensions will be the solution of
> $$\arccos(\text{ratio} ) = \kappa\pi +  \sqrt{1 - \text{ratio} ^2}.$$
> In higher dimensions, it will look like a spherical cap. For other shapes of vicinities, we may also derive the effective radii according to Theorem 3.4, though the specific forms may vary.
>
> **Regarding the comment: “How do the upper and lower bounds in this work relate to the concept of certified radius, which is a significant measure of robustness in various probabilistic robustness methods, such as Randomized Smoothing?”**
>
> That would be a really interesting and insightful question to discuss. The following might not be a solid theoretical answer, as we need to properly work out the answer. Intuitively and in my personal opinion, in the optimal condition of the probabilistic robustness, we will know that those points with probabilistic robustness will also have a vicinity for deterministic robustness. If we assume the optimal probabilistic robust accuracy is $a$ for some distribution given $\epsilon$, then we can conservatively assume that those $1-a$ points are not guaranteed with deterministic robustness and their radii are 0. If we conservatively assume that no points earn a larger certified radius. As a result, we will know that the certified robustness in terms of the radius notion (the one you suggested) would most likely be lower bounded by $(a\epsilon(1 - (2\kappa)^{\frac{1}{n}}))$

---

### Official Review · Reviewer_mx9G · 2024-11-07

**Soundness:** 3
**Presentation:** 3
**Contribution:** 2
**Rating:** 6
**Confidence:** 3

**Summary:**

The submission studies the Bayes error of probabilistic robustness. The notion is that robustness is confirmed if the wrong prediction probability is smaller than some threshold. The submission theoretically analyzed the Bayes error under this notion, derived a necessary condition for the Bayes optimal classifier, and revealed its relation with deterministic robust accuracy and the upper bound of vanilla accuracy. Empirical results show that voting increases probabilistic robust accuracy.

**Strengths:**

1. Comprehensive and novel theoretical analysis of probabilistic robustness notion from the Beyes error perspective. The analysis reveals that the Bayes-optimal classifier has the voting property (Theorem 3.2), which is of practical value and systematically connects probabilistic robustness with deterministic robustness and vanilla accuracy.

2. The presentation is good. Content is well organized and easy to follow.

**Weaknesses:**

1. The analysis largely inherits the notion and methodology from [Zhang and Sun, CAV 2024]. Technical novelty is not very significant.

2. The presentation can be made more self-contained: (1) It would be good to show the formal definition of deterministic robustness and how the Bayes optimal classifier is derived in Eqn. (7). (2) It would be good to explain how the direct method from Ishida et al can be used to compute the Bayes error of real-world datasets.



Minor:
1. Line 350: vicninty -> vicinity
2. Figure 2: what do $b_a, b_p, b_d$ mean? Some explanation preferred.

**Questions:**

Line 364, why does "$n$ grows" make the probabilistic robust accuracy higher? In my understanding, with larger dimensionality, achieving the robustness under the same Linf radius would become harder.

---

> ### Author Response · Authors · 2024-11-20
> **Response to Reviewer mx9G**
>
> Thank you for your insightful review and feedback. We hope to address your concerns and answer your questions below:
>
> **Regarding the comment: “W1.The analysis largely inherits the notion and methodology from [Zhang and Sun, CAV 2024]. Technical novelty is not very significant.”**
>
> Thank you for your comments. We would like to clarify that our work is much more than merely applying the same analysis from [Zhang and Sun, CAV 2024] to a different metric.
>
> First, the task in this study can be more challenging. Due to the difference between the definition of deterministic robustness and probabilistic robustness, establishing an upper bound for probabilistic robustness requires knowing the proportion of adversarial examples, which demands quantitative analysis in each vicinity and makes probabilistic robustness more difficult to analyze theoretically.
>
> Second, the method that we used to establish the upper bound is different. [Zhang and Sun, CAV 2024] used convolution to establish the upper bound of deterministic robust accuracy. In contrast, we primarily adopt directional derivative (*e.g.*, Theorem 3.4, Corollary 3.5) and higher-dimensional geometry (*e.g.*, Theorem 3.4, Corollary 3.6).
>
> Thirdly, we establish more than just the upper bound for probabilistic robustness. We quantitatively give the vicinity size of deterministic robustness that probabilistic robustness can ensure in the optimal condition. Particularly, this connection, denoted by $\epsilon(1 - (2\kappa)^{\frac{1}{n}})$ in Corollary 3.6) explains why even a small $\kappa$ can lead to high probabilistic robust accuracy.
>
> **Regarding the comment: “The presentation can be made more self-contained: (1) It would be good to show the formal definition of deterministic robustness and how the Bayes optimal classifier is derived in Eqn. (7). (2) It would be good to explain how the direct method from Ishida et al can be used to compute the Bayes error of real-world datasets.”**
>
> Thanks for the constructive comment and we will revise the draft accordingly. In particular, the following definition of deterministic robustness has been added in the draft as Eq (1).
> $$P_{(\mathbf{x}, \mathbf{y})\sim D} ((h(\mathbf{x}) = \textnormal{y}) ) \land \forall\boldsymbol{x'}\in\mathbb{V}(\mathbf{x}).~ h(\boldsymbol{x'}) =  h(\mathbf{x}) )$$
>
>
> We have also added a brief introduction to the derivation of  [Zhang and Sun, CAV 2024] in Appendix D and how to the derivation from [Ishida et al. 2023] in C.1.
>
> **Regarding the comment: “Line 350: vicninty -> vicinity”**
>
> Thanks for pointing that out. We have fixed it in the revised version.
>
> **Regarding the comment: “Figure 2: what do $b_a$, $b_p$, $b_d$  mean? Some explanation preferred.”**
>
> We will expand the caption of Figure 2 to make it clear: “Comparing the SOTA classifier performance with upper bounds, *i.e.*, 1 - the Bayes error of vanilla accuracy ($b_a$), probabilistic robust accuracy ($b_p$), and deterministic robust accuracy ($b_d$).”
>
> **Regarding the comment: “Line 364, why does "$n$ grows" make the probabilistic robust accuracy higher? In my understanding, with larger dimensionality, achieving the robustness under the same Linf radius would become harder.”**
>
> Thanks for your constructive questions. We agree that a large $n$ often correlates to lower deterministic robustness. Yet, according to $\epsilon(1 - (2\kappa)^{\frac{1}{n}})$ in Corollary 3.6, we show that when the vicinity size for probabilistic robustness is fixed and $n$ grows, the size of vicinity size in which deterministic robustness is guaranteed drops quickly. This is a reverse effect of the curse of dimensionality. Intuitively, since guaranteeing deterministic robustness in a much smaller vicinity is easier, the presented upper bound of probabilistic robust accuracy likely grows.
>
> Once again, we thank Reviewer for the time and effort in helping improve our paper. Should further clarification be required, feel free to ask, and we would be happy to respond.

---

> > ### Comment · Reviewer_mx9G · 2024-11-26
> >
> > Thanks for your response. Most of my concerns are addressed. However, I still feel that the technical novelty is limited, and the Beyes error properties of probabilistic robustness may not be of concern to a broader audience. Hence, I raised the score slightly to 6.

---

> > > ### Author Response · Authors · 2024-11-26
> > >
> > > Thank you for reading our rebuttal and for updating the review!

---

### Official Review · Reviewer_8ryv · 2024-11-09

**Soundness:** 2
**Presentation:** 3
**Contribution:** 2
**Rating:** 3
**Confidence:** 5

**Summary:**

The paper investigates probabilistic robustness in neural networks, proposing it as a feasible alternative to deterministic robustness against adversarial samples. Using Bayes error as a foundation, the study demonstrates that probabilistic robust accuracy can achieve a higher bound than deterministic robust accuracy due to reduced uncertainty. Theoretical results reveal that the accuracy bound for probabilistic robustness improves as κ increases, validated empirically through experiments on various datasets.

**Strengths:**

1. The paper is well-structured, with a clear and logical flow throughout.
2. The mathematical derivations are thorough, and the background definitions are comprehensive, making it easier for readers to grasp the problem the authors aim to address.

**Weaknesses:**

1. The primary concern is that the problem addressed in this paper lacks significance, which in my opinion limits its acceptance. The main contribution is the derivation of certified accuracy bound from a Bayes perspective, but the paper does not provide effective insights into improving model robustness—a key focus in this field. Although the authors propose a robustness error bound through the lens of Bayes Error, they do not attempt to use this perspective to develop a method that could enhance model robustness, which remains the core issue in this area.

2. The conclusions regarding the bound are not particularly surprising. Given that randomized smoothing allows for bound failure, the probabilistic robust accuracy is inherently higher than deterministic robust accuracy, rendering this result somewhat expected.

**Questions:**

Since Bayes error refers to the minimum achievable error for any classifier on a given problem, does this imply that it is merely a theoretical assumption and that training a true Bayes classifier may be impractica? If so, does this mean that the derived bound serves primarily as an analytical perspective on robustness rather than offering practical experimental guidance?

---

**Post-Rebuttal Comments**

Since I can no longer provide further comments at this stage, I am submitting my final response here.

After reading the rebuttal, I think Issue 1 still remains. The authors have not explicitly identified the technical challenges addressed in their proofs, and I still find the proofs insufficiently solid.

Regarding Issue 2, I would first like to clarify that PAC bounds are also model-agnostic and hold uniformly for classifiers within the hypothesis class. Additionally, the authors did not address my Questions 6-7. In my view, the method used by the authors to estimate the Bayes error is the same as that of Ishida et al., 2023, and the paper does not propose its own estimation method tailored to the task of certified robustness.

For Issue 3, the authors misunderstood my question. The bounds provided in this paper are for probabilistic certified robust accuracy, while the current methods capable of providing probabilistic certified accuracy are those based on randomized smoothing (noting that this involves adding Gaussian noise during prediction, not just at training time). The baselines compared in the paper do not provide certified robust accuracy during prediction, so I find the experiments problematic. Moreover, the idea of voting (Table 1) is precisely the concept used in randomized smoothing, where Gaussian noise is added during prediction for voting.

For these reasons, I maintain my score.

---

> ### Author Response · Authors · 2024-11-20
> **Response to Reviewer 8ryv (Part 1/2)**
>
> We thank the reviewer for the thoughtful feedback and valuable suggestions.
>
> **Regarding: “The primary concern is that the problem addressed in this paper lacks significance, which in my opinion limits its acceptance. The main contribution is the derivation of certified accuracy bound from a Bayes perspective, but the paper does not provide effective insights into improving model robustness—a key focus in this field. Although the authors propose a robustness error bound through the lens of Bayes Error, they do not attempt to use this perspective to develop a method that could enhance model robustness, which remains the core issue in this area.”**
>
> Thank you for your comments. We believe our work is significant for the following reasons.
>
>
>
> * Our work provides the first method to compute the upper bound for probabilistic robust accuracy. Knowing the upper limit is beneficial in many ways, *e.g.*, it provides a way of measuring whether a certain technique has reached the optimal performance, or whether a method is too good to be true (if its performance is beyond the upper bound) [Ishida et al, 2023].  Also, a well-derived upper bound helps evaluate a new solution, as an alternative to the relative evaluation against current best solutions [Theisen et al., 2021].
> * Our work indeed also provides guidelines on how to design practical methods for improving robustness. Specifically, Theorem 3.2 establishes that under the optimal condition for probabilistic robustness, each prediction matches the majority prediction in the vicinity. This suggests that a practical method for probabilistic robustness should probably consider voting during inference. We empirically verify this in the first research question in the experiment. (Line 427, does voting always increase probabilistic robust accuracy empirically?)
>
> * Our work also contributes to the understanding of probabilistic robustness. We derive, in the optimal case of probabilistic robustness, how much deterministic robustness will be simultaneously guaranteed. This builds a formal link between probabilistic robustness and deterministic robustness and lays the groundwork for future theoretical and practical research on robustness.  We show that a small relaxation in $\kappa$ may bring a large relaxation in effective vicinity size where deterministic robustness is guaranteed. In practice, this may help a user choose $\kappa$, or choose whether to adopt probabilistic robustness, according to the specific security requirements.
>
> **Regarding the comment: "The conclusions regarding the bound are not particularly surprising. Given that randomized smoothing allows for bound failure, the probabilistic robust accuracy is inherently higher than deterministic robust accuracy, rendering this result somewhat expected."**
>
>
>
>
>
> Thank you for your comments. We agree that the qualitative relationship between the upper bounds of deterministic robust accuracy and probabilistic robust accuracy might be expected indeed. To actually derive a quantitative relationship is however much more challenging. Our work makes it possible to answer questions such as what is the exact upper bound of probabilistic robust accuracy, and how the tolerance level kappa affects the upper bound. Furthermore, we are able to precisely explain why the upper bound of probabilistic robust accuracy is much higher than that of deterministic robust accuracy, *i.e.*, a non-zero kappa would cause the effective vicinity size to shrink exponentially, especially when dimension n is large, as shown in $\epsilon(1 - (2\kappa)^{\frac{1}{n}})$.

---

> ### Author Response · Authors · 2024-11-20
> **Response to Reviewer 8ryv (Part 2/2)**
>
> **Regarding the comment: “Since Bayes error refers to the minimum achievable error for any classifier on a given problem, does this imply that it is merely a theoretical assumption and that training a true Bayes classifier may be impractical? If so, does this mean that the derived bound serves primarily as an analytical perspective on robustness rather than offering practical experimental guidance?”**
>
> Thank you for your comments. Kindly find our response as follows.
>
> First, in many situations, approaching the Bayes classifier is practically possible. The work of Ishida et al. [2023] shows that many image classifiers have already achieved an accuracy very close to the upper bound.
>
> Second, as discussed above, even if it is challenging to develop a true Bayes classifier, knowing the upper bound is still beneficial in many ways. In brief, as discussed by Varshney et al. [2020], Theisen et al. [2021], and Ishida et al. [2023], an upper limit can practically be a reference for the performance of a new solution.
>
> Lastly, as discussed earlier in the first answer, while our derived bound primarily serves as an analytical perspective on robustness, the derivations and findings (e.g., Theorem 3.2) may still offer practical experimental guidance. This could aid both potentially improving robustness and deciding the kappa value according to the specific security requirements.
>
>
>
> ###### Lav R. Varshney. “Addressing Difference in Orientation toward Competition by Bringing Fundamental Limits to AI Challenges”. In: NeurIPS workshop, ML Competitions at the Grassroots (CiML 2020). Dec. 2020.
>
> ###### Ishida, T., Yamane, I., Charoenphakdee, N., Niu, G., & Sugiyama, M. Is the Performance of My Deep Network Too Good to Be True? A Direct Approach to Estimating the Bayes Error in Binary Classification. In The Eleventh International Conference on Learning Representations.
>
> ###### Theisen, R., Wang, H., Varshney, L. R., Xiong, C., & Socher, R. (2021). Evaluating state-of-the-art classification models against bayes optimality. Advances in Neural Information Processing Systems, 34, 9367-9377.

---

> ### Comment · Reviewer_8ryv · 2024-11-29
> **Follow-up questions**
>
> Thank you for the clarification and sorry for the late response. After reassessing the contributions of the paper, I find that the proofs do not seem to involve significant challenges, which makes it difficult to justify the paper’s acceptance. To better understand the contributions of the paper, could you clarify what specific challenges have been addressed in this work and where the difficulties in the proofs?
>
> 1. Lemma 3.3: Since $\mu_k(x)$ (Eq. (14)) represents a probability, the shift $\mu_k(x+\phi \hat{\phi}) - \mu_k(x)$ naturally relates to the integral of the changes in the PDF after shifting (Eq. (42)). The proof of Lemma 3.3 appears to involve straightforward computations regarding probabilities based on Eq. (42) and does not seem to present significant challenges. Could you explain where the difficulty lies?
>
> 2. Theorem 3.4: The paper attempts to show that deterministic robustness is a weaker condition than probabilistic robustness. This conclusion appears evident, as probabilistic robustness inherently allows for an error $\kappa$, making it a more relaxed condition. As stated on line 362, "with a much smaller vicinity," the proof also seems straightforward, relying on $\phi_1$ and $\phi_2$. Could you clarify what makes this proof challenging? Additionally, demonstrating the tightness of the bound would enhance its value.
>
> 3. Corollary 3.5: What is the motivation behind this corollary? I did not find it directly used in the following proofs. Why is it necessary to analyze the case where $\delta \to 0$?
>
> 4. Theorem 3.7: This theorem appears to rely heavily on Zhang & Sun (2024) and seems to address the same conclusion as Theorem 3.4, namely that deterministic robustness results in a smaller vicinity. Could you clarify if there are significant differences between the two results?
>
> 5. Corollary 3.8: Similar to Theorem 3.4, this result also seems evident, as deterministic robustness is inherently a weaker condition than probabilistic robustness. The proof appears similar to that of Theorem 3.2, which involves analyzing monotonicity w.r.t. $\kappa$ based on error differences $e(\kappa_1)-e(\kappa_2)$. Could you explain the novelty or challenges in this result?
>
> 6. Experiment: Regarding the code in “Training for probabilistic robustness.ipynb”, what is `305` in the line:
> ```
> x__ = x_.repeat(305, 1, 1, 1)
> ```
>
> 7. Based on line 1227, the Bayes error calculated in this paper seems to rely directly on the algorithm from Ishida et al. Does the paper propose a new method for estimating the Bayes bound, or is it dependent on existing work?
>
> 8. According to the code, the voting mentioned in the paper involves adding uniform noise to the input $x$ to enhance the model's robustness. This idea is the same as randomized smoothing (Cohen et al., 2019), which is essentially a majority voting scheme, except that RS uses Gaussian noise. There has been extensive research on RS. Could you clarify the setting used in this paper for comparison with RS? Does the RS baseline include adding Gaussian noise? Or is Gaussian noise added first, followed by uniform noise?
>
> 9. In my understanding, what should theoretically be bounded is the probabilistic certified robust accuracy. Why is it that the baseline methods DA and CVaR, as provided in their original papers, only report empirical robust accuracy rather than certified robust accuracy? These two methods appear to be variations of adversarial training, and as far as I understand, only RS can provide probabilistic certified robust accuracy. Could you clarify this point?

---

> ### Comment · Reviewer_8ryv · 2024-12-02
>
> As the reviewer response period is about to end and the authors have not addressed my questions, I have adjusted my score based on my current understanding of the paper. The paper has three main issues:
>
> 1. As a theoretically focused paper, its major problem lies in the lack of technical depth in the proofs, which makes me question whether the addressed problem is genuinely challenging and worth solving. As I pointed out in Questions 1-5, I have been trying to identify whether the theorems overcome some significant difficulties, such as relaxing key conditions or employing innovative mathematical tools. However, most of the computations appear to involve basic applications of probability theory.
>
> 2. The problem addressed in this paper does not seem particularly important. Certified robust accuracy, as a lower bound for empirical robust accuracy, theoretically means that any upper bound for empirical robust accuracy can also serve as an upper bound for certified accuracy. There are already many works on bounding empirical robust accuracy in adversarial PAC area. In other words, the paper focuses on a relatively small problem within certified robustness, which is not particularly compelling.
>
> 3. If the theoretical contributions are not strong, I would place greater emphasis on the experimental part of the paper. Unfortunately, the experiments also seem problematic. As I mentioned in Questions 6-9, the paper claims to propose an upper bound for probabilistic certified accuracy. However, the baselines compared seem to be adversarial training algorithms, and the specific implementation of randomized smoothing, the only method capable of providing certified robustness, is unclear. This makes it difficult for me to determine whether the claimed improvements represent a fair comparison. I recommend including more RS algorithms for comparison, such as the baselines discussed in [1].
>
> Thus, I have adjusted my score and confidence level accordingly.
>
> [1] Confidence-Aware Training of Smoothed Classifiers for Certified Robustness, Jeong et al., AAAI 2023.

---

> > ### Author Response · Authors · 2024-12-03
> >
> > **Regarding "As a theoretically focused paper, its major problem lies in the lack of technical depth in the proofs, which makes me question whether the addressed problem is genuinely challenging and worth solving. ... However, most of the computations appear to involve basic applications of probability theory."**
> >
> > Thank you for your comments. It may be hindsight to say the qualitative relationship between the upper bounds of deterministic robust accuracy and probabilistic robust accuracy is not surprising. Instead, our work makes the first step to providing sound, clear, and detailed proof of how probabilistic robust accuracy is upper bounded and *therefore* how this upper bound relates to those of vanilla accuracy and deterministic robust accuracy.
> >
> > Specifically, in Lemma 3.3, $\mu_k$ has a condition term and a $v$ term in the integral but till Eq (42) the condition term is eliminated. Theorem 3.4 gives proof that in the best case, benign neighbours will form a solid core around the tested input (rather than to other extremes, *e.g.*, spread as uniformly as possible). Corollary 3.5 directly follows Theorem 3.4, proving that even when the vicinity approaches zero, the change in $\mu_k$ is still bounded by some real number. This is mostly for theoretical completeness. Theorem 3.7 and Corollary 3.8 are formal conclusions naturally derived from preceding lemmas and theorems. Although Theorem 3.7 could be interpreted as a proposition, its strong relevance to the studied topic justifies itself as a theorem [[Math120](http://sporadic.stanford.edu/Math120/writing.pdf)]. Clearly showing these results, even when intuitive and sometimes step-by-step, enhances the reliability of the work.
> >
> >
> > **Regarding "The problem addressed in this paper does not seem particularly important. ... In other words, the paper focuses on a relatively small problem within certified robustness, which is not particularly compelling."**
> >
> > Thank you for your comments. The PAC robustness bound primarily refers to the error bound subject to the model used [Huang et al., 2022], which is fundamentally different from the upper bound studied in this work. The bound derived in this study is regarding the task-wise performance itself which is only subject to data distribution [Ishida et al., 2023] it is the inherent error in data that is model-agnostic.
> >
> > Deterministic robustness might often lead to compromised accuracy, and probabilistic robustness can overcome the trade-offs between the accuracy and the sample complexity of worst-case and average-case learning, which may be deemed as a more feasible solution. Given its importance, knowing its upper bound can help in measuring if a technique approaches optimal performance, detecting unrealistic results (performance exceeding the bound) [Ishida et al., 2023], and evaluating new solutions independently of current best methods [Theisen et al., 2021].
> >
> >
> > ###### Pei Huang, Yuting Yang, Minghao Liu, Fuqi Jia, Feifei Ma, and Jian Zhang. 2022. $\epsilon$-weakened robustness of deep neural networks. In Proceedings of the 31st ACM SIGSOFT ISSTA.
> >
> > **Regarding "If the theoretical contributions are not strong, I would place greater emphasis on the experimental part of the paper. ... This makes it difficult for me to determine whether the claimed improvements represent a fair comparison. I recommend including more RS algorithms for comparison, such as the baselines discussed in [1]."**
> >
> > The baselines we used include CVaR [Robey et al., 2022], and RS training [Cohen et al., 2019], and neither are (limited to) adversarial training algorithms. Robey et al. [2022] define the probabilistic robust classifiers and probabilistic robust accuracy concepts. Their [Robey et al., 2022] method effectively addresses probabilistic robust accuracy, as it is designed to optimize this property. RS training adds Gaussian noise to training samples, which generally improves the robustness without using adversarial search or any particular adversarial examples. These are fair and promising approaches to realise probabilistic robustness. Besides, improving probabilistic robustness is a separate problem from the one studied in this work. This research question aims to offer an empirical observation of Theorem 3.2, which is not our main contribution.
> >
> > Thank you for recommending the article. We will consider adding it to our final version.

---

### Meta-Review · Area_Chair_L4uZ · 2024-12-23

**Metareview:**

This paper investigates the probabilistic robustness with focus on deriving an upper bound of probabilistic robust accuracy through bayes error. The finding and derivation on the monotonic relation between probabilistic accuracy upper bound and tolerance level is interesting but not surprising. There are some discussions in the rebuttal regarding the theoretical and experimental results, topic of interest, and significance of the results. In my opinion, this topic is of interest to the robust ML community, however the current form of the paper is not qualified yet for publications, with a few major recommendations to further strengthen the paper: (1) overcomplicated presentation of the main results in Sec 3, the authors are suggested to present the most important results and make the structure easy to follow. Current form contains too many branches and requires reorganization. (2) missing connection to probabilistic robustness literature and baselines, specifically the probabilistic certified robustness / verification in the neural network literature. (3) experiment result: randomized smoothing belongs to deterministic robustness instead of probabilistic robustness, the incorporation of RS in the experiments is questionable and confusing. The authors are also encouraged to take into account the reviewers suggestions/confusions in the rebuttal period to further improve the clarity and strength the results of the paper.

**Additional Comments On Reviewer Discussion:**

Besides the above unresolved 3 major points, other shared concerns seem to be well-addressed by the authors. This includes:
(i) the difference between recent work (Zhang & Sun, 2024) and this work: the authors clarified that (Zhang & Sun, 2024) only focused on deterministic robustness
(ii) the significance of topic to the ML community: the importance and advantage of probabilistic robustness over deterministic robustness

---

### Decision · Program_Chairs · 2025-01-22

Reject